# Spatially-optimized urban greening for reduction of population exposure to land surface temperature extremes

Emanuele Massaro [1] ✉, Rossano Schifanella[2,3], Matteo Piccardo[4], Luca Caporaso[1,5], Hannes Taubenböck [6,7], Alessandro Cescatti[1] & Gregory Duveiller [1,8]

The population experiencing high temperatures in cities is rising due to anthropogenic climate change, settlement expansion, and population growth. Yet, efficient tools to evaluate potential intervention strategies to reduce population exposure to Land Surface Temperature (LST) extremes are still lacking. Here, we implement a spatial regression model based on remote sensing data that is able to assess the population exposure to LST extremes in urban environments across 200 cities based on surface properties like vegetation cover and distance to water bodies. We define exposure as the number of days per year where LST exceeds a given threshold multiplied by the total urban population exposed, in person·day. Our findings reveal that urban vegetation plays a considerable role in decreasing the exposure of the urban population to LST extremes. We show that targeting high-exposure areas reduces vegetation needed for the same decrease in exposure compared to uniform treatment.

Over half of the world's population lives in cities[1], and thus in localized hotspots covering less than 3% of the Earth's land surface[2]. Moreover, the number of urban dwellers is predicted to grow by an additional 2.5 billion by 2050[3]. The large and increasing share of the urban population had stimulated renewed attention to studies that investigate the livability of urban environments[4]. Urban areas are places where humans have altered their local environment and local climate in the most radical way, replacing the natural land cover and vegetation with impervious materials that have lower albedo and heat capacity[5], substantially reducing the cooling effect of water evaporation[6]. Urban ecosystems can be described by having fundamentally three different types of elements: (i) impermeable surfaces, (ii) vegetated surfaces, and (iii) water bodies. The spatial integration of these three elements determines the physical properties of the urban landscape surface, thereby affecting both the microclimate of cities and the frequency of extreme heat[7–9].

The combination of global warming and the urbanization process is ratcheting up the number of humans exposed to health-endangering heat. This exposure has tripled in recent decades, a faster rise than previous research suggested[10]. As a consequence, heat waves in urban climates have a profound impact on humankind as the climate warms up[11] and this will increase in the future[12]. Regional studies confirm that the population is foreseen to be increasingly exposed to weather-related hazards and, in particular, to heat waves[13] leading to an ever-higher risk of fatalities[14,15]. Important aspects of psycho-physical well-being and quality of life crucially depend on the climatic conditions of the environment in which people live[16]. Changes in the global temperature can generate a wide range of consequences for human health. Direct consequences are linked to the bioclimatic well-being of people, whereas indirect consequences are linked to the complex interactions between environmental conditions and the spread of infectious[17] and allergic

[1]European Commission, Joint Research Centre (JRC), Ispra, Italy. [2]University of Turin, Turin, Italy. [3]ISI Foundation, Turin, Italy. [4]Collaborator of the European Commission, Joint Research Centre (JRC), Ispra, Italy. [5]National Research Council of Italy, Institute of BioEconomy (CNR-IBE), Rome, Italy. [6]German Aerospace Center (DLR), Munich, Germany. [7]University of Würzburg, Würzburg, Germany. [8]Max Planck Institute for Biogeochemistry, Jena, Germany. ✉e-mail: emanuele.massaro@ec.europa.eu

diseases[18]. The positive effects of climate-related human well-being are many[19]: from the reduction of the risk of numerous chronic diseases in adulthood (such as diabetes and cardiovascular conditions, obesity, and asthma), to the acceleration of recovery after surgery, to the reduction of hospital admissions and premature mortality, up to better pregnancy outcomes and improved mental health[20]. To achieve an intelligent and sustainable development of the urban fabric, there is the necessity to move away from the traditional approach of territorial planning that ignores social and environmental factors[21]. Instead, there is the need to work together with a common goal, using shared methodologies and regulations to handle the impacts of climate change on city growth and development[22]. To manage the rapid increase of these direct and indirect climate risks in urban environments, there is a need for knowledge-based policies that foster the design, application, and monitoring of adaptation plans. For this scope, new research efforts are needed to (i) increase the understanding of the specific causes of extreme heat, and to (ii) offer monitoring and modeling systems of the urban thermal environment at the required spatial and temporal scales. To cope with the increasing need to foster climate mitigation and adaptation, it is therefore important to have spatially and temporally detailed information, possibly at a sub-daily temporal frequency. So far, the lack of adequate data has actually prevented the development of numerical methods to derive appropriate metrics and diagnostics to fully describe, or profile, these factors. The increasing availability of time series of thermal remote sensing information from satellite platforms is now supporting effective quantification of urban heat consistently across the globe and with high spatial detail[23-25]. Most previous studies on the subject focused on the Urban

Heat Island (UHI) phenomena, or more specifically Surface Urban Heat Island (SUHI) when studied within the remote sensing field[23]. SUHI is defined as the difference between the urban and the rural land surface temperature (LST), and it occurs because the highly dense artificial materials of city cores heat up considerably stronger than their rural surroundings (see Fig. 1). Depending on the type of surface surrounding the city, the value of SUHI can be either positive or negative (which in the latter case is referred to as an urban cool island or UCI). When cities are surrounded by vegetated areas (see Fig. 1B), the magnitude of the SUHI can be seen as a proxy of the rural vegetation density instead of a measure of the thermal comfort of an urban environment. To make a step-change in the approach to the problem, the design and evaluation of urban adaptation plans should be city-specific and based on the detailed assessment of the city heat characteristics[26].

To assess the latter requirements, in this study, we set a spatial model to predict the absolute value of the exposure of the population to LST extremes. We present an approach based on a spatial regression model to predict the exposure of urban populations to LST extremes. Despite its simplicity, the model is able to assess the exposure of the urban population to LST extremes using only remote sensing observations. We measure exposure as the number of days and nights when LST exceeds a certain threshold, multiplied by the number of people affected: this is a surrogate version of population exposure to extreme heat using air temperature[10,27]. The spatial regression model uses greenness and proximity to water bodies as predictors and has been tested in 200 cities across various climates and has consistently proven its accuracy. Finally, we utilize our findings to assess the impact of urban greening initiatives in reducing the exposure of urban

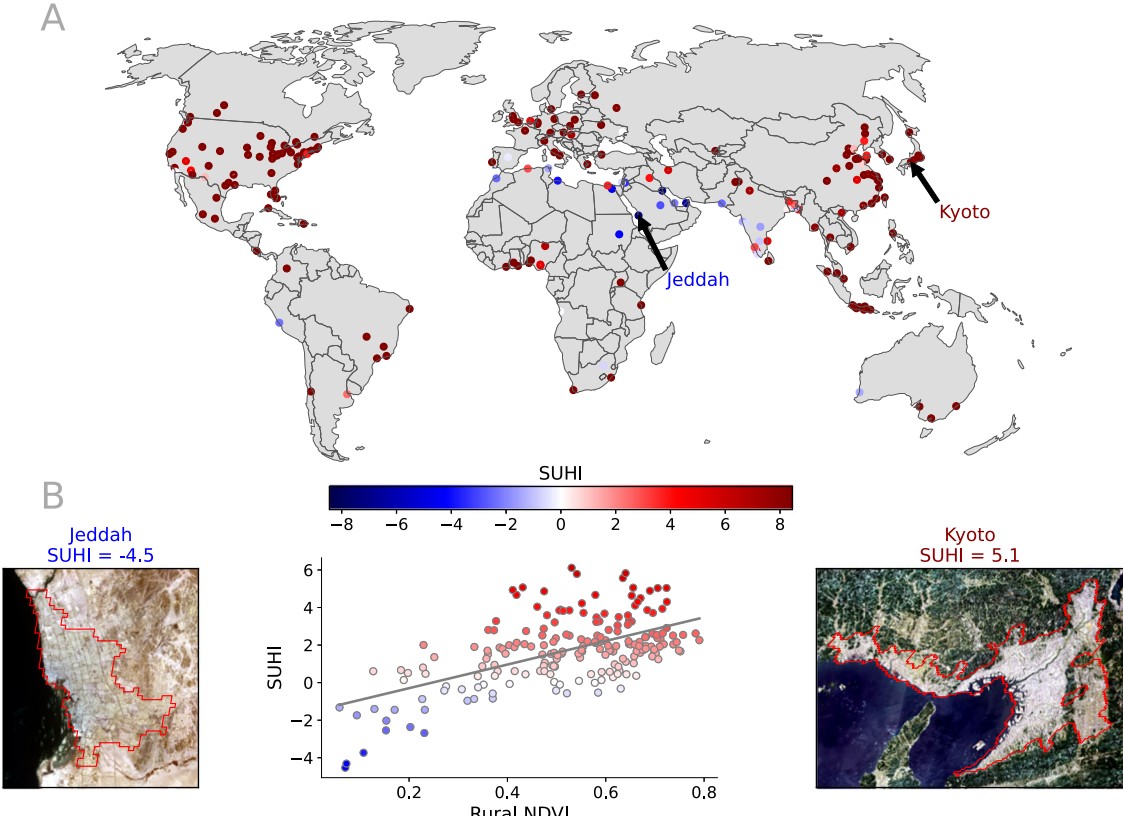

**Fig. 1 | Surface Urban Heat Island (SUHI). A** Averaged value of SUHI between 2010 and 2020 for the 200 cities analyzed in this research. **B** As an example, we report the opposite cases of Kyoto and Jeddah that have positive and negative values of the SUHI, respectively. The satellite images show the difference in the vegetation of the adjacent rural areas. On the right we show the correlation between SUHI and the average value of the rural greenness expressed with the Normalized Difference Vegetation Index (NDVI) as defined in the Methods section for all the cities. See data availability and code availability for publicly available dataset and codes for generating the figures.

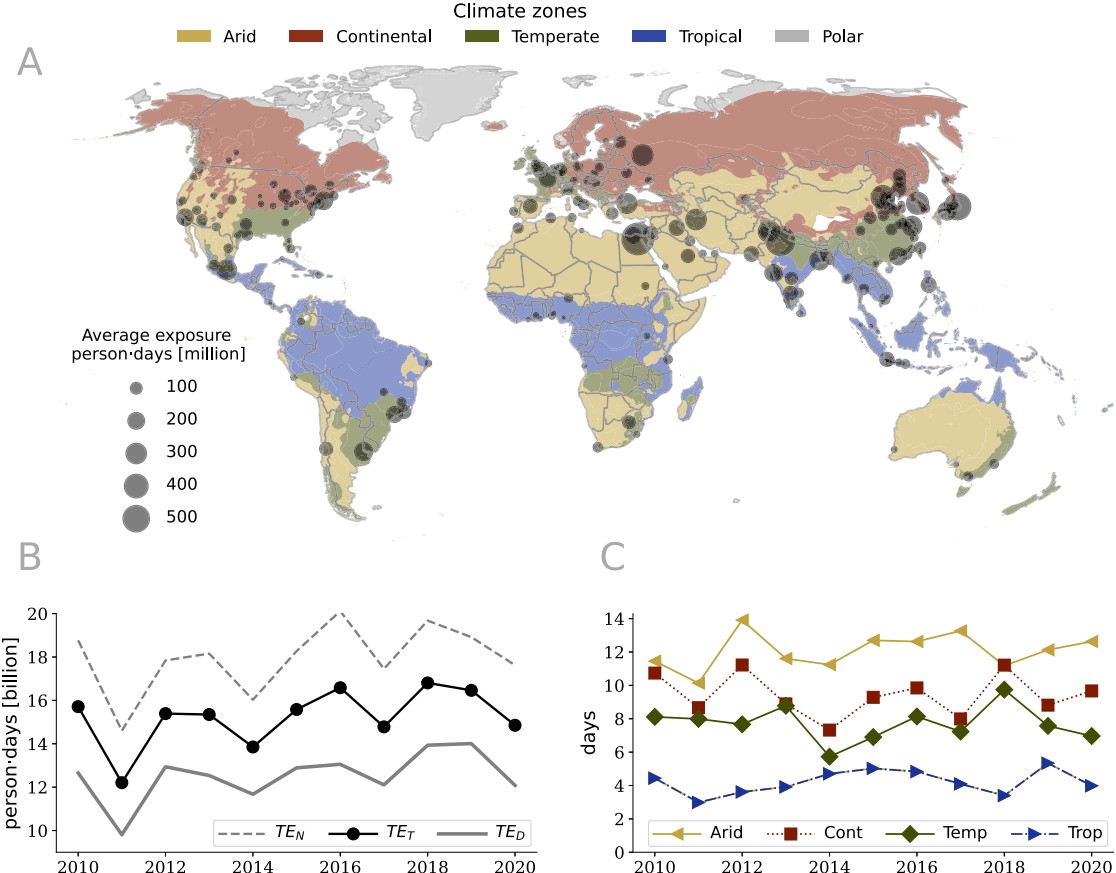

**Fig. 2 | Population exposure to Land Surface Temperature (LST) extremes.**
**A** Population exposure in the cities within the five climate zones defined by the Köppen-- Geiger climate classification system[62] in mean number of person-days per year. The size of the dots corresponds to the population exposure averaged over the 10 years of the observations. **B** Values of the exposure over the years. **C** Average value of the exposure divided by the population that corresponds to the average number of days and nights over the thresholds for each climate zone (in the legend, Cont stays for Continental, Temp for Temperate and Trop for Tropical). See data availability and code availability for publicly available dataset and codes for generating the figures.

populations to LST extremes. Our research presents a globally validated model and we believe that the same methodology can also be utilized for specific, practical, and localized solutions.

## Results

We focus the analysis on 200 cities worldwide that are characterized by a wide range of climatic conditions. These are divided equally per major climate zone: Arid, Continental, Tropical, and Temperate (see Fig. S1 in SI). The urban environment is described by a 1 km spatial resolution multiband raster, with several variables, where for each pixel and for each year, we have aggregated various types of information, namely: daytime and nighttime values per summer where LST exceeds a threshold, average values of the Normalized Difference Vegetation Index (NDVI), distance to water surfaces, and population (see Materials and Methods section and Fig. S2 in SI). We apply a spatial regression model to predict the exposure of the urban population to LST extremes. In line with the recent literature, we define the urban population exposure $TE$ as the number of days per year where LST exceeds a threshold, called exposure threshold, multiplied by the number of people, expressed in persons·days[27], where days correspond to the average between daytime and nighttime values over thresholds. For each urban environment, we define the daytime/nighttime exposure threshold as the 90th percentile of the temporal distribution of the daytime/nighttime LST over 20 years (from 2000 to 2020), as shown in Figs. S3 and S4 in SI. By this definition it is possible to compare the results obtained in geographic areas with a different climate and in different seasons of the year[28], and the 90th percentile

ensures an effective definition of LST extremes, where the outliers are not taken into account as suggested by the Intergovernmental Panel on Climate Change (IPCC)[29] and the World Meteorological Organization (WMO)[30]. For each pixel in the images of the cities, we calculate the number of daytime and nighttime values that exceed exposure thresholds for the 3 warmest months of the year, as illustrated in Fig. S5 in the SI for Paris. To account for the daily cycle, we define the total exposure $TE_T$ as the average of the daytime exposure $TE_D$ and the nighttime exposure $TE_N$, i.e., $TE_T = (TE_D + TE_N)/2$, as demonstrated in Fig. S6 for Paris. In Fig. 2 we show the values of $TE_T$ across space and time: in Fig. 2A we show the average values for each city in the different climate zones. In Fig. 2B we show the trends of the value of the exposures over time. We can see a clear increment of the urban population exposure to LST extremes from 2010 to 2020 that is mostly caused by the urbanization process as observed in Fig. 2C where we show that the average exposure divided by the population is almost constant over the years for the different climate zones.

### Urban greening to reduce the exposure of the urban population to LST extremes

Before presenting the findings, it is necessary to elaborate on urban greening, including its benefits and limitations. Vegetation, including trees and other plants, has a significant cooling effect on urban environments, making it a simple and effective way to reduce the exposure of urban populations to extreme heat[19,26]. Vegetation can lower surface and air temperatures by providing shade and increasing evaporation[31]. Shaded surfaces can be up to 20−45 °F (11−25 °C) cooler

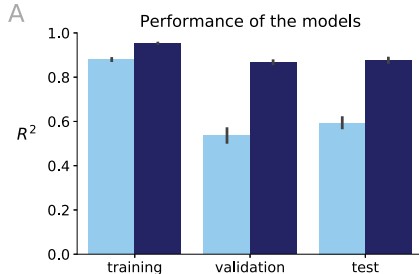

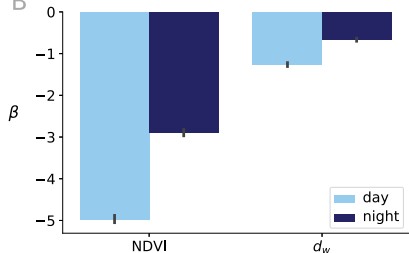

**Fig. 3 | Spatial lag models for the estimation of the daytime ($TE_D$) and the nighttime ($TE_N$) exposures. A** Performance of the models in terms of $R^2$ and (**B**) the coefficients of the models for the Normalized Difference Vegetation Index (NDVI) and distance to water bodies ($d_w$). See data availability and code availability for publicly available dataset and codes for generating the figures.

than unshaded surfaces[32]. Evapotranspiration from vegetation, either alone or in conjunction with shading, can help mitigate peak summer temperatures[33,34], whereas urban surfaces covered in concrete and asphalt contribute to the opposite effect. Due to these biophysical effects, urban vegetation is an important tool in climate-smart urban planning. Urban vegetation design is a crucial adaptation strategy that reduces heat stress by modifying the surface properties of cities. However, there is still a lack of dedicated modeling tools to aid in the design of city-specific plans based on the level of intervention required to achieve climate targets. For instance, microscale models that simulate the local impacts of urban design and land use on the urban microclimate and the potential for microclimate interventions to mitigate urban heat exposure are needed. While proximity to urban green spaces has advantages for health and cooling[19,35], cities also face competition for space and resources[36,37]. In this study, we focus solely on measuring urban greening to reduce the exposure of urban populations to LST extremes. However, this is not the only solution to urban heat exposure, and other solutions, such as increasing the albedo of surfaces[38], are known to have positive effects. In practical applications, urban greening can effectively reduce land surface temperatures[39], however, tradeoffs with water resources must be considered[40].

**Limitations and advantages of the proposed approach**
In this study, we quantify the exposure of the urban population to LST extremes by measuring the number of days that LST exceeds a certain threshold, multiplied by the number of people exposed: this definition is a surrogate version of exposure to extreme heat using air temperature[10]. We acknowledge that using thermal remote sensing data as a proxy for population exposure to heat in urban environments has its own limits[41]. LST may not fully represent the thermal environment experienced by people in urban areas as it measures the temperature of the surface rather than the air[41,42]. In addition, LST measurements can be affected by the urban canyon effect[41,42], and errors can arise due to the complex dependence of emissivity on urban materials and vegetation[43]. Recent literature has highlighted the need to assess thermal exposure at a hyper-local level, considering factors such as exposure to the sun, wind, and relative humidity[44]. For these reasons, in this research, we do not focus on the population exposure to heat but on urban areas with high values of LST. Despite these limitations, the study of LST provides advantages for research purposes, including a more detailed and global representation of temperature patterns[45] that allows for inter-city comparisons at a fine temporal (daily) and spatial (1 km resolution) resolution that would not be feasible with air temperature data. Our study focuses on global patterns and both seasonal and longer-term trends, averaging values of day and night LST at a spatial resolution of 1 km over the 3 warmest months of the year for a period of ten years[41,45]. We do not consider hourly or daily conditions at specific locations, where the differences between air and surface temperatures might be considerable due to rapid changes in weather-scale processes[41,45]. Our methodology can be adopted and reproduced for any city in the world due to the availability of consistent satellite information captured globally with the same resolution, methodology, and acquisition time every day. In this study, we only consider the effect of urban vegetation as a mitigation strategy for population exposure to LST extremes, while other strategies such as promoting energy efficiency and regulating building codes[46] or increasing the albedo of urban surfaces[47] have not been considered.

**The spatial linear regression model**
We present a simple and global model to predict the number of days ($Y_d$) and nights ($Y_n$) over thresholds for each city where $Y_d/Y_n$ correspond to the average number of days/nights over the threshold observed from 2010 to 2020. The model uses as predictors two local land surface properties: $X_1$ the fraction of green area within a pixel (as represented by the Normalized Difference Vegetation Index or NDVI); $X_2$ distance to water bodies $d_w$ as described in the Methods section. Given the presence of a clear spatial autocorrelation of the variables (see Table S1 in SI), we adopted a spatial linear regression model called spatial lag model (SLM)[48]. We studied two models for both $TE_D$ and $TE_N$: we use the the Google Earth Engine (GEE) platform[49] to compute the days where LST is greater than the thresholds (day and night). To assess the performance of the spatial model, we compute $R^2$ (see Fig. 3 and Fig. S8 in SI) and mean absolute error (MAE) within a k-fold cross-validation setting (see Methods and Fig. S7 in SI for details). In Fig. 3(A) and in Fig. S8 in SI we show that, despite its simplicity, the model is able to predict $TE_D$ and $TE_N$ averaged over the 3 warmest months of the years in the 200 cities with high accuracy. The result shows that an $R^2 \geq 0.8$ in the test-set within a k-fold cross-validation setting is obtained. Our analysis shows that the SLM performs significantly better than a Ordinary Least Squares (OLS) regression model as shown in Fig. S9 reporting negative low values of $R^2$ in the training phase and negative values during the validation phase, underlying the importance of accounting for the spatial correlation. Finally, model inferences suggest that the magnitude of the exposure during the warmest months of the year are largely controlled by urban vegetation with average regression coefficients ($\beta_{NDVI}$, $\beta_{d_w}$), calculated as the mean value of the best coefficients estimated during the training/validation steps of the k-fold cross validation (see Fig. S7 in SI), are (−4.7,−1.2) for $TE_D$, and (−2.7,−0.6) for $TE_N$. It is noteworthy that the coefficients related to vegetation $\beta_{NDVI}$ and distance to water bodies $\beta_{d_w}$ are negative, indicating that an increase of the surface covered by green areas and water bodies within cities would result in a reduction of the exposure to LST extremes of the areas that are affected by such an intervention.

**The mitigation strategy through vegetation increment**
We use the SLM with the values of the coefficients computed during the cross-validation setting to estimate the vegetation increment required for reducing the exposure of the urban population to LST

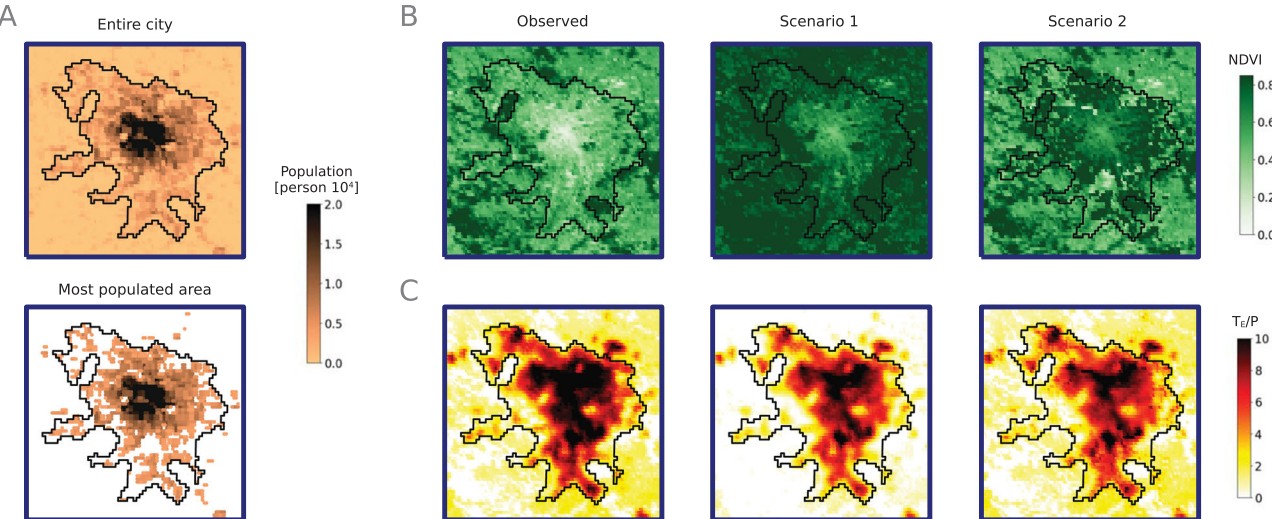

**Fig. 4 | Exposure reduction for the city of Paris. A** We show the number of inhabitants at each pixel of the entire city and the most populated areas. **B** We present the values of the Normalized Difference Vegetation Index (NDVI) for each observed pixel and for three different scenarios. The Observed scenario corresponds to the observed values of the NDVI and the corresponding measured exposure. In Scenario 1, the value of the NDVI is increased homogeneously at all pixels by 0.3. In Scenario 2, the value of NDVI is increased only at the most populated areas of the city to achieve the same exposure reduction as in Scenario 1. **C** We show the value of the total exposure divided by the population for each pixel in the different scenarios. See data availability and code availability for publicly available dataset and codes for generating the figures.

extremes as a function of the NDVI (see Methods section), while keeping the values of the distance to the water bodies fixed to the observed values. It should be noted that the vegetation increase is here applied without taking into account any urban or social considerations (e.g., space availability, financial means, energy efficiency, etc), as these are out of the scope of what we want to do in this particular work. In practice the considerations with respect to mitigation strategies for a specific city need to be analyzed in detail, as there are, for example, limits in how much a pixel can be greened up without compromising its capacity to contain living and functional places. In Fig. 4 we show the results for the city of Paris as an example. There we simulate $TE_T$ when NDVI is increased by 0.3 with respect to the observed values, to the maximum value of 0.85. We apply this increment for all the pixels that have an observed valued of the NDVI < 0.85 of the city and we quantify the exposure reduction as $\Delta TE_T(\%) = (TE_T^1 - TE_T^0)/TE_T^0$, where $TE_T^0$ and $TE_T^1$ are respectively the values of the total exposure before and after the NDVI increment. Figure 4(A) shows the entire city as well as these pixels where 80% of the entire population live respectively where the black boundary corresponds to the urban boundary as defined by the Global Human Settlement Layer (GHSL) dataset[50] and the blue boundary is the urban environment as defined in the Methods section and in Fig. S2 in SI. Figure 4(B) and (C) show the value of the NDVI and the total exposure divided by population for three different scenarios. The values of the NDVI and the total exposure for the different scenarios are reported in Table S2 and in Fig. S10 in SI. We show the observed values of the NDVI that lead to a total exposure $TE_T^0 \sim 164$ million person·days. Scenario 1 corresponds to the value of the NDVI increased homogeneously at all pixels that leads to a total exposure $TE_T^1 \sim 144$ million person·days. We show that by increasing the NDVI homogeneously by 0.3, it is possible to reduce the exposure of urban population to extreme heat by ~12%. We also show that by increasing the NDVI homogeneously, we count an increment of 44% of the entire NDVI of the city (see Table S2 and Fig. S10 in SI). In Scenario 2 we estimate the NDVI increase in just the most populated areas which would lead to the same exposure reduction of ~12%. This reduction could be achieved with a 0.38 increase of the NDVI in the corresponding pixels, shown in Fig. 4(B), and it corresponds to a 14% NDVI increment over the whole image (see Table S2 and Fig. S10 in SI). With this

simulation we show that it is possible to optimize the vegetation increment by targeting specific areas of the city.

In a subsequent analysis, we make simulations in which we increase NDVI for all 200 cities considered in this study as reported in Fig. 5. We show that, on average, increasing the NDVI by 0.3 results in exposure reductions of 12%, 13%, 16%, and 32% for cities located in temperate, arid, continental, and tropical regions, respectively (Fig. 5B). The required amount of NDVI increase per pixel was computed for the most densely populated areas of each city to achieve the same exposure reduction. Our findings demonstrate that targeting specific areas with higher exposure led to significant differences in NDVI increments. On average, 90%, 20%, 26%, and 33% of the NDVI could be saved for arid, continental, temperate, and tropical regions, respectively (Fig. 5C). The average local NDVI increment for the most populated areas is approximately 0.45 to optimize the NDVI increment in the urban environment (Fig. 5C). We also considered different NDVI increments ranging from 0.1 to 0.5. Figure 5(E) illustrates the exposure reduction for various NDVI increments. Results indicate that exposure linearly decreased with NDVI increments, but stable patterns of relative NDVI savings were observed as shown in Fig. 5(F). The relative NDVI savings were calculated using the equation NDVI(%) = 100(NDVI80 − NDVIall)/NDVIall, where NDVIall and NDVI80 represent the sum of NDVI values after interventions on the entire city and on the most populated areas, respectively. On average, it was possible to save up to 70% of greening by targeting specific areas to achieve the same exposure reduction as random homogeneous interventions as shown in Fig. 5F. In Fig. 5G we show the difference between local and global NDVI increments and we observe that an average NDVI increment of ~0.15 could achieve the same exposure reduction for different global increments.

## Discussion

In this study, we set a spatial lag model (SLM) to reproduce the exposure of the urban population to LST extremes for 200 cities worldwide: we defined exposure as the number of days per year where LST exceeds a heat exposure threshold multiplied by the total urban population exposed, in person·days: this definition can be seen as a surrogate version of population exposure to extreme heat using air temperature[10,27]. Our approach, which is exclusively based on remote

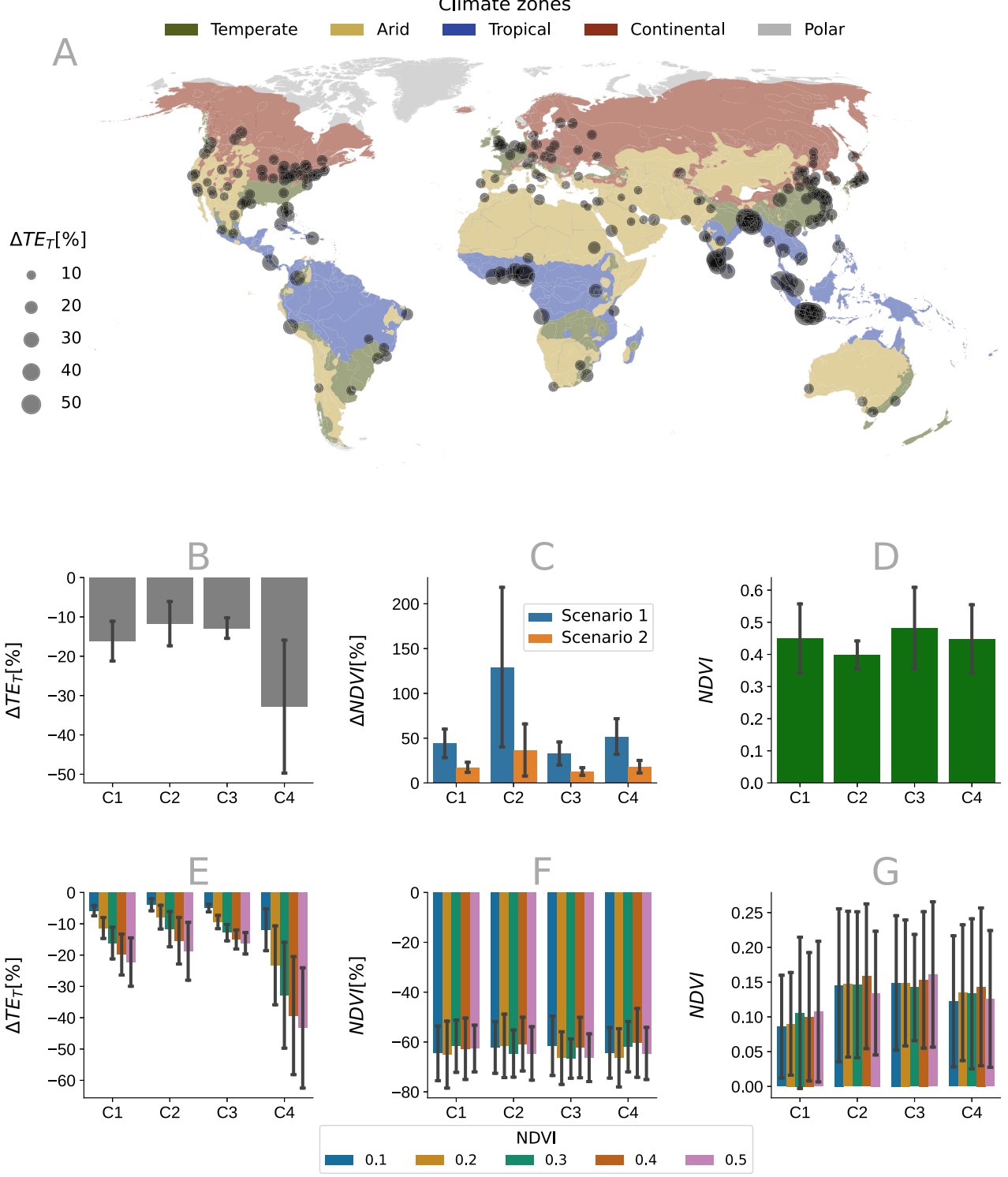

**Fig. 5 | Exposure reduction for all cities across the different climate zones.** The *x*-axis displays the four climate zones: Temperate, Arid, Continental, and Tropical (C1, C2, C3, and C4, respectively). **A** Show the values of the total exposure reduction ($\Delta TE_T(\%)$) for all cities with an increment of the Normalized Difference Vegetation Index (NDVI) of 0.3. **B** We show the aggregated values of $\Delta TE_T(\%)$ for the different climate zones. **C** We compare the global NDVI increment required to achieve exposure reduction by targeting the entire city versus the most populated pixels. Scenario 1 shows a global NDVI increment to achieve exposure reduction by targeting the entire city, while in Scenario 2, NDVI is increased only in the most populated pixels as described in Fig. 4. **D** We present the average and standard deviation of the local NDVI increment for Scenario 1 to achieve the same exposure reduction with a homogeneous increment of 0.3. **E** The plot displays the average and standard deviation of the exposure reduction for different values of NDVI increment for all cities in the different climate zones. **F** We show the relative NDVI savings achieved by targeting the most populated areas for different values of NDVI increment. **G** We present the difference between the local and global NDVI increments for different values of global NDVI increment. See data availability and code availability for publicly available dataset and codes for generating the figures.

sensing data, is able to predict the population exposure to LST extremes of urban environments. We have analyzed the impact of the vegetation change in reducing the exposure of urban population to LST extremes and computed this value for each pixel of each city. This provides theoretical options for already highly dense urban areas, often characterized by a lack of space and housing. Lastly, it should be mentioned again that our approach is large-scale and in some ways generalizes patterns across broad geographic and climatic regions[45]. Even if the SLM has been tested in cities across various climates and has consistently proven its accuracy, climate adaptation strategies need to be implemented locally, e.g., how additional green spaces are distributed, what types of greenery are planted, details that, for space constraints, are not part of this study.

We also make clear that is not simply a matter of increasing the total amount of urban greening in a city, but where the greening is targeted. While in this paper we focused on the most populated areas of the city, in the next steps we suggest to focus on targeting other relevant parts of the urban environments. It has been shown that the effects of urban greening can vary depending on the location, with irrigated landscapes in arid regions lowering nighttime temperatures only in the least vegetated neighborhoods[51]. In addition, there are multiple studies that have highlighted the relationship between land surface temperature (LST) and historically marginalized areas within cities[52]. It is important to also target interventions in more vulnerable areas[53]. Although the marginalized areas of cities may not be part of the current study, the inequity of where vegetation is currently located should be addressed in the introduction and interpretation of the results to provide a more comprehensive understanding of the issue[54]. Moreover, exposure to heat is not limited to the outdoors and that people may also be exposed to heat within buildings. The availability of indoor cooling, as well as the gradient in air conditioning between wealthy and poor neighborhoods, is a crucial factor to consider when assessing exposure to extreme heat[55]. These considerations emphasize the requirement for a deeper analysis of a multi-dimensional approach when thinking about exposure to extreme heat, and the significance of considering the built environment and socio-economic factors in assessing vulnerability: this is a complex topic that deserves further investigation.

In summary, this research used a spatial lag model to assess and predict with high accuracy the exposure of urban population to extreme heat and it reveals quantitative results that give decision-makers concrete guidance and general options to develop climate adaptation strategies in planning the urban landscape. In this study, we utilized Land Surface Temperature (LST) to assess exposure to extreme heat, considering both daytime and nighttime observations to account for the diurnal temperature exposure. While there may be some limitations in using LST for estimating population exposure at specific spatial and temporal scales[41], previous studies have shown that LST and air temperature are closely correlated over extended periods of time[45], such as the 3-month period over a ten-year span we analyzed in this research. In addition, this method can easily be replicated by anyone for any city worldwide due to the availability of consistent satellite data that is captured globally at the same resolution, using the same techniques, and obtained at almost the same time every day and every night[56]. Our research presented a globally validated model, and we believe that the same methodology can also be applied to specific, practical, and localized solutions. Finally, we employed our findings to assess the efficacy of urban greening initiatives in reducing extreme heat exposure for urban populations. Our results demonstrated that targeting areas of the city where the population is more exposed to extreme heat leads to a substantial reduction in the amount of vegetation coverage needed to achieve the same decrease in exposure compared to a uniform treatment. As a next step, in order to achieve even greater improvement, it would be beneficial to use urban greening in combination with other

mitigation strategies such as increasing surface albedo[47] and water bodies[57].

## Methods

### Data and key variables

This study makes use of the Land Surface Temperature (LST) and Normalized Difference Vegetation Index (NDVI) products from the Moderate Resolution Imaging Spectroradiometer (MODIS) instrument on board of the NASA's Terra and Aqua platforms: all data have been collected from the Google Earth Engine (GEE) platform[49]. For each sample city and its relative urban boundary, the average data for the 3 warmest months of the year from 2010 to 2019 were collected and extracted from GEE as data points in a table. The urban boundary, the population layer and land use (i.e., presence of water bodies) was taken from the EU Global Human Settlement Layer (GHSL)[58]. To define the 3 warmest months for each year and city we rely on the ERA5 Copernicus data https://cds.climate.copernicus.eu/. All the data have been aggregated at spatial resolution of 1km in the Mollweide projection.

**Land surface temperature and heat thresholds.** LST is from the MOD11A1 V6 product that provides daily LST and emissivity at $1\,km$ nominal spatial resolution. We used the the Google Earth Engine (GEE) platform to compute the values of the number of days and nights over thresholds. For each city, the day and night thresholds have been computed as the 90th percentile of the distribution of the LST over 20 years of observation (from 2002 to 2022) of all pixels of the city as shown also in Fig. S3 and Fig. S4 in SI. This product provides daily per-pixel Land The acquisition time of the MOD11A1 Version 6 is around 10:30am and 10:30pm respectively for day and night.

**Normalized difference vegetation index.** NDVI is from MOD13Q1 V6 product at 250 m nominal resolution and defined as:

$$NDVI = \frac{NIR - Red}{NIR + Red} \tag{1}$$

where *NIR* and *Red* are the atmospherically corrected bi-directional surface reflectances (masked for water, clouds, heavy aerosols, and cloud shadows) measured at $250m$ nominal spatial resolution in the near-infrared and red wavebands respectively (see MODIS product description[56]).

**Distance to water bodies.** The third predictive variable of the model is an indicator of each point to the closest water body. Information on water presence was retrieved from the Global Human Settlement Data Layer produced by the European Commission Joint Research Center for the period 1984–2015[59]. The water occurrence is a measurement of the water presence frequency (expressed as a percentage of the available observations over time actually identified as water). The provided occurrence accommodates for variations in data acquisition over time in order to provide a consistent characterization of the water dynamic over time. The variable used in our model is defined as:

$$d_w = \frac{1}{D_w^2} \tag{2}$$

where $D_w$ is the distance of each pixel to the water surface. In the model, we used the normalized value of $d_w$ between 0 and 1 in order to make this value in the same range as the other predictors. We used the Guidos toolbox[60] to estimate the distance to all pixels to water bodies worldwide.

**Definition of cities and urban boundaries.** The urban boundaries are set to the areas provided by EU GHSL. In particular, we used the Degree of Urbanization, a new global definition of cities, urban and rural areas

dataset available online (https://ghsl.jrc.ec.europa.eu/degurba.php). The city boundaries are defined taking a rectangular buffer of $5\,km$ around the GHSL urban boundaries, as shown in Fig. S2 in SI as an example for the city of Guangzhou.

**Data aggregation.** The workflow to generate an aggregated dataset with all the necessary information starts with the definition of the urban boundary as defined above. Once the urban boundary has been defined, we divided the urban environment into a $1\,km \times 1\,km$ grid as shown in Fig. S2 in SI. In each cell of the grid, we then aggregated the information we collected. Thereafter, all the information necessary for running the model and performing the analysis has been spatially merged and measured over the grid. For the population, we counted the total population living in each cell. For the LST we assigned each data point to its relative cell, for NDVI we assigned several points to a single cell given the higher spatial resolution and we computed the average, while for the distance to the water bodies ($D_w$) we computed the distance from the epicenter of the grid to the closest water body point. We used the Guidos toolbox[60] to compute the distance of each pixel to the closest water body as defined by the GHLS[58].

**Climate aggregation**
Cities are classified based on their climate zone following the Köppen climate classification which is one of the most widely used climate classification systems. The Köppen climate classification divides climates into five main climate zones, with each zone being divided based on seasonal precipitation and temperature patterns. The five major climate zones are then further divided into subzones. The second letter indicates the seasonal precipitation type, while the third letter indicates the level of heat. We define five main clusters as shown in Fig. S1.

**The spatial model**
In this research we make use of the so called Spatial Lag Model (SLM) that primarily addresses spatial autocorrelation[48] (see also Table S1 SI). Spatial autocorrelation refers to the case when the dependent variable exhibits a non-random pattern over our spatial units after controlling for other covariates. Positive autocorrelation reflects value's similarity in space, and negative autocorrelation reflects value's dissimilarity in space. In a matrix notation the SLM reads as (see SI for a detailed description of the SLM):

$$Y = \beta X + \rho W_y Y + \varepsilon \qquad (3)$$

where $Y = \beta X + \varepsilon$ corresponds to a standard linear regression model, while $W$ is the spatial weighting matrix applied to the observed variable, $Y$, together with a spatial autoregression parameter, $\rho$ that reflects the spatial dependence inherent in our sample data, measuring the average influence on observations by their neighboring observations[61]. In a spatial lag model the dependent variable among our neighbors influences our dependent variable. The spatial weights matrix, $W$ is standardized such that its rows sum to 1, hence it is effectively including a weighted average of neighboring values into the regression equation. The neighbors can be identified by different methods, such as distance based methods (k-nearest neighbors, distance specified) and contiguity methods (queen, rook)[48]. In this research we use a k-nearest neighbors approach with $k = 8$ where all the neighbors are equally weighted.

## Acknowledgements

The study was funded by the Exploratory Project Thermopolis of the European Commission, Joint Research Centre. R.S. acknowledges partial support from the European Union's Horizon 2020 innovation action project Go Green Routes under grant agreement No. 869764.

## Author contributions

E.M., G.D., and A.C. designed the research; E.M. performed the research, collected and analyzed the data; E.M., G.D. A.C., R.S., L.C., M.P., H.T. revised and discussed the results of the research, wrote, and revised the paper.

## Competing interests
The authors declare no competing interests.

## Data availability
All the data used in this research is available online. Data for LST and NDVI can be accessed in the Earth Engine Data Catalog (https://developers.google.com/earth-engine/datasets). The water and population data can be retrieved from the Urban Global Settlement Layer catalog (https://ghsl.jrc.ec.europa.eu/dataToolsOverview.php) of the European Commission, Joint Research Centre (JRC). All the data collected and generated from this study are available online from https://doi.org/10.5281/zenodo.7848332 or on request from the first author.

## Code availability
All the codes in order to generate the figures, run the models and perform all the analysis (in the form of Python notebooks and scripts) have been collected and shared in a public repository https://doi.org/10.5281/zenodo.7848332 or on request from the first author.

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
