## [Peer Review File · Nature Communications]

Spatially-optimized urban greening for reduction of population exposure to land surface temperature extremesREVIEWER COMMENTS

Reviewer #1 (Remarks to the Author):

“Measuring intervention strategies to reduce the exposure of urban population to extreme heat” by Massaro et al.

This submission combines spatially explicit land surface temperature (LST), land cover, and population data to assess the potential for LST reduction via urban greening in populated areas across the main climate zones globally using a spatial regression model. In general it is well executed and presented, and there is potential to explain the results more clearly. However, while this submission is scientifically interesting, its practical significance is in doubt. The authors do not discuss the disadvantages of LST for practical purposes. I have suggested some references for the authors’ consideration, and I suggest the authors better communicate the uncertainty in terms of the practical relevance of their findings, especially in the abstract and conclusions.

NOTE: No line numbers were provided, so I’ve described the relevant section/paragraph for each of my comments.

ABSTRACT, and elsewhere: the term “heat exposure” is used – what is meant by this? Land surface temperature typically won’t relate very closely to the actual exposure to longwave radiation or air temperature by a pedestrian. See Stewart et al. (2021) for example and their idea of incomplete (pedestrian) surface temperature. Moreover, in the biometeorological literature actual heat exposure is calculated (and related to far more than longwave radiation, but also shortwave radiation/shade, air temperature, wind speed, humidity). Also, “thermoscapes” based only on 2-D surface temperatures (i.e., roofs, roads, treetops, etc) is a limited definition. What about air temperature, shade, etc?

Stewart, I.D., Krayenhoff, E.S., Voogt, J.A., Lachapelle, J.A., Allen, M.A. and Broadbent, A.M., 2021. Time evolution of the surface urban heat island. *Earth's Future*, 9(10), p.e2021EF002178. I also strongly suggest the authors read the following:

Martilli, A., Roth, M., Chow, W.T., et al. 2020. Summer average urban-rural surface temperature differences do not indicate the need for urban heat reduction.

And this article, which they already cite, contains detailed discussion on several issues related to the use of LST that the authors should consider in my view, but currently do not:

Voogt, J.A. and Oke, T.R., 2003. Thermal remote sensing of urban climates. *Remote sensing of environment*, 86(3), pp.370-384.

In other words, while satellite-derived LST is readily available and therefore often used in these kinds of analyses, I don’t think it is the appropriate data to answer the scientific questions posed here. The link between LST and actual exposure of urban residents is extremely tenuous at best. Moreover, for assessment of impacts of tree, in particular, it is not appropriate. The major impact of trees in terms of heat exposure is shade, and LST cannot capture the impacts of shade.

INTRODUCTION

First paragraph:

- Cities typically have lower albedos than rural areas. Some typical urban materials often have lower albedos, e.g. asphalt, many roofing types.
- Frequency of extreme heat is mostly controlled by synoptic weather patterns, not by the composition of the urban landscape.

Third paragraph:

- LST does not effectively capture the effects of shade from trees, since shade reduces temperatures of surfaces under trees that are not accessible to satellite-based remote sensors; therefore, it is not clear why the authors discuss tree shade.

METHODS

- Distance to water bodies: the distance will be much more relevant if the water is upwind (for the dominant wind direction) than downwind; I presume this is not taken into account?
- Time of day is not reported – what time of day are the results representative of? This should be provided in the Results section as context. How well is the day-to-day temporal variation over the summertime sampled?

RESULTS

Climate and population exposure to extreme heat

- I disagree that “exposure of the urban population to extreme heat” is calculated. It is the co-location of high LST with population. High roof temperature, a primary contributor to LST, is unlikely to impact the population except perhaps those living on the top floor, depending on building construction. Also, the “warmest areas of the city” may be quite different if air temperature were the metric. Finally, “extreme heat” is usually used to indicate synoptically-driven heat (e.g. heat waves), or at very least experiencing (air) temperature exceeding 90th+ percentile of the *temporal* distribution of temperature. The 95th percentile of temperature spatially is not very relevant if nowhere in the city is too hot.

Reduction of the population exposed to extreme heat

- What exactly is meant by “the population exposed to the warmest areas of the urban environments can be reduced by 50%”? Does it mean, for the example of Paris (threshold of 34.4 C), that temperature is reduced to 34.35 C? The issue with the use of percentages above or below a threshold is that a small reduction in temperature that nevertheless crosses a defined threshold (e.g., from 34.5 C to 34.3 C for Paris) is not that relevant or impactful. Even the significance of a 1 C LST reduction for the population is not clear, both in terms of the relevance of this drop in LST and the relevance of LST to actual exposure to heat of urban residents as mentioned above. (Also note that sometimes Kelvin are used, and sometimes degrees C – suggest more consistency.)

- What is the meaning of the bar plots (e.g. Fig. 4C) by climate zone. Are these results representative of all cities in each climate zone? E.g., are the chosen cities representative of all cities in that climate zone? This is perhaps a point for discussion.

Temperature reduction

- Figs. 5B and S14 are unclear and would benefit from more explanation. Why is the Local vegetation reducing?

CONCLUSIONS

- Delete the final 2 sentences.

Reviewer #2 (Remarks to the Author):

This study explores the potential exposure of urban populations to extreme heat under different greening scenarios. It relies on an interesting approach of combining data about urban population and satellite-derived LSTs. The authors claim that increasing the overall urban vegetation by 3% can already lead to reduction in exposure to extreme heat of 50%. I think that this statement may be technically correct for a certain definition of exposure. However, I think that this definition of exposure may not be that useful when trying to better understand how to protect the urban population from the harmful effects of heat in urban areas. I also think that the results are presented in a way that suggests an easy solution to reducing the detrimental effects of heat extremes in cities, but the effects of vegetation in cities are not that clear and easily quantifiable, especially when looking more holistically at how the effects of vegetation vary over time (e.g. diurnal cycle) and space. I think that the manuscript may in its current form also lead to misinterpretations of the potential benefits of climate change mitigation (i.e. atmospheric CO₂ reduction) in comparison to climate adaptation (i.e. more vegetation in cities). I am suggesting major revisions to improve the current version of the manuscript.

Major comments:

Before mentioning my main concerns, I have to say that I am not 100% sure how exposure to extreme heat and the reduction of it is defined. I think I could not explicitly find it in the methods nor in the Supplementary Information. If I have not missed it, I would like to ask the authors to specify this more clearly in the manuscript. My understanding is (based on the first paragraph of

the results and Figure S2) that exposure to extreme heat occurs (as per this definition) when LST is higher than the 95th percentile (as defined within one city) based on average(?) LSTs and I am assuming that exposure is "reduced" whenever temperatures fall below the temperature for the 95th percentile as predicted by the statistical model for an increase in NDVI.

If I am correct about the definition of exposure, I see certain limitations regarding exposure and the potential to reduce it for the urban population when increasing vegetation cover (NDVI). I think these limitations should be discussed in the manuscript and I would suggest to make a less generalized claim than saying that an increase of 3% of vegetation can reduce exposure to extreme heat by 50%.

1.) There is now quite an extensive body of literature and controversy about the usefulness of LST in studies about the urban heat island and the reduction of urban heat (e.g. Martilli et al. 2020). While I would not say that LST data is useless, I think that we should be aware of certain limitations when defining exposure based on LST: LST can only serve as a proxy for air temperatures and e.g. human thermal comfort. For example, during the day the canopy urban heat island (~2m air temperature difference) is often quite small (Oke et al., 2017). One explanation seems to be (simply put) that the city shapes its own atmospheric boundary layer in such a way that heat can be transported more efficiently into higher layers of the atmosphere. This seems to be one of the reasons why air temperatures (during the day) over heavily sealed and high-rise building areas are sometimes not much higher than over vegetated land outside of the city. This can be problematic for two reasons: (1) The spatial pattern of the canopy urban heat island (i.e. air temperatures) may overall (but oftentimes not) correspond to the pattern of LSTs. This means the areas exposed to the highest LST (>95th) may not always be the ones exposed to the highest air temperatures. I think this should be discussed, since we may argue that air temperature matters often much more. (2) While the LST distribution ranges between, e.g., 25 and 35 °C, the air temperature in cities during the day is often not varying spatially by more than 0-2°C. Assuming that a heatwave in a changing climate may be, e.g., 2°C hotter (compared to some baseline), this may mean that the exposure to extreme heat and the 95th percentile should have a much broader definition than it has now (including a time dimension)? I think that this should be discussed and I think it would be great to have a much broader discussion in the paper why the author's think that their definition of exposure is useful and not merely arbitrary. For example, why is exposure defined per city and not globally, why not with a fix temperature threshold etc.

2.) The diurnal cycle is also something crucial to consider. My understanding is that the MYD11A1 product has nighttime and daytime temperature in it, but that only daytime was used. During nighttime, the effects of vegetation can of course be quite small. Some studies even suggest that, e.g., trees could cause warming (depending on the urban setting). Exposure to extreme heat in this study is thus very narrowly defined as exposure to extreme heat during the day. However, the exposure to extreme heat during the night is at least equally if not more relevant (for example for human health).

3.) The areas with the highest population density may often be somewhere in or close to the city center and with a lot of high-rise buildings. There are two things that come to my mind that should probably be considered and discussed: (1) While the MODIS satellite/sensor will often see overall high LSTs for these areas, there are some places within these areas where temperatures are actually low. For example, in deep street canyons there is often a lot of shade. In these canyons the effect of vegetation is often relatively small (e.g. trees will not provide additional shade). Vegetation may actually not have that much of an effect here and of course (especially during night) trees may reduce cold air flows into the cities because they are natural obstacles. This may be especially relevant in densely built areas with a small amount of wind corridors. (2) When thinking of exposure to extreme heat, we could think of pedestrians and how they are exposed to heat. I think it is probably a valid assumption that there are a lot of pedestrians in highly populated areas and because of this many pedestrians may be exposed to heat (unless they walk in heavily shaded deep canyon streets etc.). However, another dimension is that people are exposed to heat within buildings. In buildings there is often air conditioning. So the number of people that (and during how much time of the day) are exposed to extreme heat, may depend very much on the availability of air condition. There might be strong gradients in air conditioning between wealthy and poor neighborhoods (from the city center to the outskirts). Even though I am

not that familiar with the topics of exposure and vulnerability, it seems to me that these things would matter and that it may even become an issue of social justice in which neighborhoods it makes sense to increase vegetation and just looking at population density may be too simple. In addition, it is of course a huge topic, whether vegetation can even have an influence on temperatures within buildings and how much vegetation effects are only occurring very close to the vegetation itself. So assuming that a grid cell would profit (the whole grid cell) from a temperature reduction is often not correct, since only the areas with vegetation will experience a significant temperature decrease, but the urban areas close to these vegetated areas will profit much less.

Minor comments (somewhat unordered because there was no line numbering, which makes it really hard to comment):

1. In the first paragraph it says "with higher albedo". I think it is more often argued that cities have a lower albedo or that it is very dependent on the building materials deployed when constructing cities in different parts of the world.

2. First page, last paragraph: "Changes in the global, ..., and allergic diseases". I don't think that this fits very well here. Maybe it can be integrated somewhere at the beginning of the paragraph or be removed.

3. Under the regression model: You say it is a multivariate regression model, but you only have one dependent variable. I am not sure that this is and should be called multivariate. At least, including multiple independent variables does in my opinion not make it multivariate.

4. I would assume that NDVI and NDBI have a strong negative correlation. How does that affect the estimated regression coefficients? I am missing a bit more information about the structure of the variables (correlation, distribution etc.) used and if there could indeed be issues with collinearity.

5. In addition, I would find it helpful to see a bit of model diagnostics. For example, how do the residuals look like? Are they normally distributed (normal q-q), what about homoscedasticity and outliers (e.g., residuals vs leverage)?

6. I have to say that I find the statistical approach overall a bit (maybe sometimes unnecessarily) complicated and sometimes the benefits are not entirely clear to me:

a) The authors are saying that: "Our analysis shows that the model performs significantly better than a standard linear regression model, underlying the importance of the spatial dimension". They also claim that they don't use any "additional climate information". I consider both statements rather misleading (but of course I may also not fully understand everything). (1) The fact that there is a large improvement when using a "spatial model" seems due to the fact that you include some form of "auto-covariate" which is constructed based on the temperature itself. In my opinion this is "additional climate information" even though it happens to be the dependent variable in the model. (2) I believe that spatial autocorrelation when analyzing LST does not matter that much. I would assume that the large performance improvement in comparison to a "standard regression" approach, would also occur, if you simply include average LST per city as an additional independent variable. I also doubt that accounting for auto-correlation is most important in terms of "performance" (R^2). Usually, I would consider it relevant to account for autocorrelation when you do some sort of statistical inference including the calculation of p-values and to a certain degree it can of course influence your regression coefficients. So to me it would make more sense to show these things (change in p-value and regression coefficients) instead of all the figures illustrating the performance.

b) I am assuming that you are using the cross-validation approach to find an optimal "rho" value (which is a hyperparameter that controls, e.g., overfitting?), but I am not entirely sure if this is described explicitly and if my understanding is correct. Wouldn't it make sense to do the whole cross-validation exercise to find an optimal rho, then estimate the regression coefficients (for a combined training and validation set) and then evaluate the error with the help of the test set? I would say that averaging the regression coefficients (betas) should not have any benefits. Do you systematically vary "rho" to find its best value? If not and if "rho" is fitted as a parameter of the

model, then I would again say that the whole cross-validation approach that you are employing does maybe not make much sense because the parameters will probably not be any better than without the cross-validation exercise.

6.) I guess the assumption that the NDVI has the same effect on temperature in different cities is more or less valid. On the other hand, I would assume that vegetation in dry regions use less water for transpiration and hence cause less cooling. Would a discussion on this make sense and did you check how much the regression coefficient of NDVI might vary across different cities?

7.) In Figure 10 I would find it more helpful to see boxplots in the left part. The right part of the figure is totally unclear to me. Probably the figure caption should be more explanatory.

8.) In the methods "Application of the model and evaluation of the mitigation strategy". The title of this section/paragraph should change. There is a lot of description about the cross-validation procedure etc. This could be an additional paragraph or be included under "The spatial model".

References

- Martilli et al. 2020. Summer average urban-rural surface temperature differences do not indicate the need for urban heat reduction
Oke et al. 2017. Urban Climates

Reviewer #3 (Remarks to the Author):

Thank you for the opportunity to review this manuscript. I applaud the authors for their selection of cities across the world, both Global North and South, and the inclusion of diverse geographies and climates. While the manuscript's methods are clearly presented, the overall framing of the manuscript and interpretation of results should be strengthened. Most critically, the manuscript currently frames land surface temperature (LST) as essentially equivalent to heat exposure. While the authors could argue LST can be used as a proxy of sorts for heat exposure, they must acknowledge the well-documented limitations of LST. This framing also influences the interpretation of the results, which need much more nuance than currently written, particularly if the paper is to be relevant to and clearly understood by decisionmakers. I offer several overarching comments first, followed by more specific line feedback.

Overarching areas of improvement:

- As stated above, a more nuanced discussion is needed in the introduction on the drawbacks and limitations of LST and the evidence of its relationship to heat exposure of community members. Community members have different heat exposure over time depending on the quality of housing, travel modes, and thermal safety at work or school, etc. If LST is to be used as a proxy for heat exposure, the authors should acknowledge the limitations and drawbacks of doing so. For instance, Krayenhoff & Voogt (2010) (below), argue that decisionmakers should aim to reduce the impacts of urban heat, not the urban heat island itself. Please refer to literature such as:
 - o Martilli, A., Krayenhoff, E.S., & Nazarian, N. (2020). Is the Urban Heat Island intensity relevant for heat mitigation studies? *Urban Climate*, 31, 100541.
<https://doi.org/10.1016/j.uclim.2019.100541>
 - o Krayenhoff, E. S., & Voogt, J. A. (2010). Impacts of urban albedo increase on local air temperature at daily–annual time scales: model results and synthesis of previous work. *Journal of Applied Meteorology and Climatology*, 49(8), 1634-1648.
 - o Kelly Turner, V., Rogers, M. L., Zhang, Y., Middel, A., Schneider, F. A., Ocón, J. P., ... & Dialesandro, J. (2022). More than surface temperature: mitigating thermal exposure in hyper-local land system. *Journal of Land Use Science*, 17(1), 79-99.
- In the introduction and discussion, I also recommend acknowledging other types of heat mitigation strategies available to decisionmakers to limit urban heat including cool/higher albedo materials and surfaces, urban design choices that can improve thermal comfort in microclimates, and the reduction of waste heat sources through the improved energy efficiency of buildings and indoor cooling as well as the reduction of gasoline-powered vehicles.
- Along the same lines, the authors should be more nuanced in the discussion of urban greening strategies for heat mitigation, and mention the critical water resource tradeoff decisions that would

need to be made in many drought-stricken areas of the world (particularly arid regions which they identify would need greater amounts of urban greening to reduce LST).

- Finally, the authors should address that it is not simply a matter of increasing the total amount of urban greening in a city, but where the greening is targeted. Gober et al. (2009) found that irrigated landscapes in an arid region lowered nighttime temperatures but only in the least vegetated neighborhoods. Similarly, there are numerous studies published about the relationship between LST and historically marginalized areas within cities (additional references below), so targeting interventions to those most vulnerable community members should also be considered. While the marginalized areas of cities are not part of your study, the inequity of where vegetation is most often already located should be addressed in the introduction and the interpretation of the results.

- o Gober, P., Brazel, A., Quay, R., Myint, S., Grossman-Clarke, S., Miller, A., & Rossi, S. (2009). Using watered landscapes to manipulate urban heat island effects: how much water will it take to cool Phoenix?. *Journal of the American Planning Association*, 76(1), 109-121.

- o Locke, D. H., Hall, B., Grove, J. M., Pickett, S. T., Ogden, L. A., Aoki, C., ... & O'Neil-Dunne, J. P. (2021). Residential housing segregation and urban tree canopy in 37 US Cities. *npj urban sustainability*, 1(1), 1-9.

- o Hoffman, J. S., Shandas, V., & Pendleton, N. (2020). The effects of historical housing policies on resident exposure to intra-urban heat: a study of 108 US urban areas. *Climate*, 8(1), 12.

- o Dialesandro, J., Brazil, N., Wheeler, S., & Abunnasr, Y. (2021). Dimensions of thermal inequity: neighborhood social demographics and urban heat in the Southwestern US. *International journal of environmental research and public health*, 18(3), 941.

Specific line feedback:

- Title: The manuscript is only focused on "urban greening strategies" versus a broader set of heat mitigation strategies, and the focus of the methods is on LST and the reduction of LST, rather than reduction of heat exposure.
- Page 1, 2nd paragraph: "As a consequence, heat waves in urban climates will have a profound impact on humankind as the climate warms up." Please reflect upon the tense in which heat waves is framed in this sentence. Heat waves are already having an impact on individuals, a risk that is increasing (versus a risk that is only in the future).
- Page 2, Paragraph 1: "To cope with the increasing need to foster climate mitigation and adaptation, it is therefore imperative to have spatially detailed, temporally sub-daily information for most, if not all cities." Not sure why this distinction is being made here, what cities do not need to cope with climate change? Recommend rephrasing to something like, "...information for cities."
- Page 2, Paragraph 1: "...subject focused on the so-called Urban Heat Island (UHI) phenomena..." Recommend removing "so-called".
- Page 3, Paragraph 1: "city thermoscape..." recommend continuing to use "urban thermoscape" as indicated in the introduction.
- Page 3, Paragraph 1: "For instance, these plans could be implemented by favouring green spaces and constructing green or cool roofs and cool pavements." This statement is correct, but in the context of the paragraph the focus is urban greening strategies, of which cool roofs and cool pavements are not part of. Recommend rephrasing to clarify the distinction or removing these strategies from this sentence.
- Page 4, Paragraph 1: "However, we are still missing dedicated modelling tools to facilitate the design of city-specific plans based on the magnitude of the intervention required to reach climate targets." Can you provide evidence from the literature that modelling tools are needed, and not training of professionals on existing modelling and information decision support tools?
- Page 4, Paragraph 2: Noting it is being referred to as "city thermoscapes" again here.
- Page 4, Paragraph 3: "It is evident that people in arid regions are located in areas of their particular city less exposed to extreme heat..." Please clarify and reconsider your interpretation of this result. The populations in arid regions may be located in areas of a city that have lower UHI relative to surrounding natural landscapes, but that does not mean they are less exposed to extreme heat.
- Page 5, Paragraph 3: "we show the amount of vegetation we need to increase in order to reduce the exposure of the urban population to extreme heat areas..." More accurate to say that the scenario shows vegetation needed to reduce the LST, which you are then using as a proxy for exposure of the urban population to heat.
- Page 5, Paragraph 3: "Whether this expansion of urban vegetation is actually feasible in that

specific area without compromising the capacity to host people should be further investigated.”
Not only the capacity of the area to host people, but also trade-offs with water use, maintenance costs of the additional vegetation, etc.

- Page 6, Paragraph 1: Although the study is focused on urban greening strategies, this would be another good place to nod to the fact that cities are not only pursuing greening, but have an additional suite of heat mitigation strategies available to them. As currently written, it sounds like the only option available to cities is urban greening and not a holistic mix of greening, cool surfaces, and waste heat reduction.
- Finally, I also recommend including line numbers in future submissions of the manuscript.

Reducing extreme heat exposure in cities through spatial analysis of urban greening

Response to reviewers

Emanuele Massaro, Gregory Duveiller, Rossano Schifanella, Matteo Piccardo, Luca Caporaso, Hannes Taubenböck, Alessandro Cescatti

Dear Reviewers,

We have revised the manuscript addressing all the comments. Please find the list of all the changes to the manuscript in our point-by-point response below. The original referees' comments are in blue. In this document, we tried to answer all the comments made by reviewers by explaining the validity of our results and methodology.

We would like to thank a lot the three Reviewers that allow us to improve significantly the manuscript and the results of the research. The major changes addressed are the following:

1. We changed the title as suggested.
2. We critically discussed the advantages and the limitations of Land Surface Temperature (LST) data for practical purposes as suggested by the three reviewers;
3. We changed the definition of population exposure to extreme heat areas, which are now defined as the number of days when heat exposure surpasses a specific threshold multiplied by the total urban population. This new definition is in line with the current literature¹.
4. In order to take into account the diurnal cycle we collected both daytime and nighttime information from the MODIS-Terra satellite and we defined daytime and nighttime thresholds. In this way we defined the number of days over thresholds as the average between daytime and nighttime values.
5. We critically discussed the scope of our work and its possible integration with other intervention strategies.
6. We added the Data Availability and Code Availability sections and we shared a public repository that contains all the data and the codes used in this research¹. All the analysis and the figures we show here are in a dedicated python notebooks called *responseToReviewers.ipynb*.

We would like to emphasize that the key contribution of this study is to establish a highly accurate Spatial Lag Model (SLM) for predicting urban population exposure to extreme heat. We then use the model's results to evaluate the impact of vegetation on reducing such exposure. We calculate exposure based on Land Surface Temperature (LST) data, which has been criticized by some studies, including Martilli et al. 2020², for its limitations in assessing the Surface Urban Heat Island (SUHI) effect. However, it's worth noting that our study doesn't focus on SUHI, which compares rural and urban LST, but rather on the absolute value of extreme heat exposure, calculated pixel by pixel. We finally show that targeting high exposure areas reduces vegetation needed for the same decrease in exposure compared to a uniform treatment.

Sincerely yours,

Emanuele Massaro, Gregory Duveiller, Rossano Schifanella, Matteo Piccardo, Luca Caporaso, Hannes Taubenböck, Alessandro Cescatti

¹https://github.com/emanuelemassaro/heat_exposure

**Reviewer 1**

This submission combines spatially explicit land surface temperature (LST), land cover, and population data to assess the poten-
tial for LST reduction via urban greening in populated areas across the main climate zones globally using a spatial regression
model. In general, it is well executed and presented, and there is potential to explain the results more clearly. However, while
this submission is scientifically interesting, its practical significance is in doubt. The authors do not discuss the disadvantages
of LST for practical purposes. I have suggested some references for the authors' consideration, and I suggest the authors
better communicate the uncertainty in terms of the practical relevance of their findings, especially in the abstract and conclusions.

We thank Reviewer 1 for appreciating our work from a scientific viewpoint and for his relevant comments that helped us to
significantly improve the manuscript. We fully agree with Reviewer 1 that we did not discuss sufficiently the limits and the
disadvantages of LST for this kind of intervention. We also agree that the practical significance of this work as it was before
could be questionable. The main goal of this research is to provide a model that is able to predict with high accuracy the
exposure to extreme heat in urban environments. We provided a global model, while models for singles cities based on spatial
regression should be developed for single cases and more in-depth practical purposes. We would like to emphasize that in the
revised version we changed the definition of exposure and we presented a geographical regression model that is able to predict
with high accuracy the number of days and nights over given thresholds. We used the parameters of the model to estimate the
increment of vegetation (NDVI) in order to reduce the exposure with a global comparison across different climate zones. Any
practical interventions should be analyzed locally for each city, and it is beyond the scope of this work, and we emphasized this
point further in the revised manuscript. In the revised manuscript we discussed these limitations in a dedicated section. In the
following, we tried to address all the comments made by Reviewer 1.

We would like to emphasize that the key contribution of this study is to establish a highly accurate Spatial Lag Model
(SLM) for predicting urban population exposure to extreme heat. We then use the model's results to evaluate the impact of
vegetation on reducing such exposure. We calculate exposure based on Land Surface Temperature (LST) data, which has been
criticized by some studies, including Martilli et al. 2020², for its limitations in assessing the Surface Urban Heat Island (SUHI)
effect. However, it's worth noting that our study doesn't focus on SUHI, which compares rural and urban LST, but rather on the
absolute value of extreme heat exposure, calculated pixel by pixel.

NOTE: No line numbers were provided, so I've described the relevant section/paragraph for each of my comments.

We thank the reviewer for this point: we added the line numbers to the revised version of the manuscript.

ABSTRACT, and elsewhere: the term "heat exposure" is used
– what is meant by this? Land surface temperature typically won't relate very closely to the actual exposure to longwave
radiation or air temperature by a pedestrian. See Stewart et al. (2021) for example and their idea of incomplete (pedestrian)
surface temperature. Moreover, in the biometeorological literature actual heat exposure is calculated (and related to far more
than longwave radiation, but also shortwave radiation/shade, air temperature, wind speed, humidity). Also, "thermoscapes"
based only on 2-D surface temperatures (i.e., roofs, roads, treetops, etc) is a limited definition. What about air temperature,
shade, etc?

Stewart, I.D., Krayenhoff, E.S., Voigt, J.A., Lachapelle, J.A., Allen, M.A. and Broadbent, A.M., 2021. Time evolution of the
surface urban heat island. *Earth's Future*, 9(10), p.e2021EF002178.

I also strongly suggest the authors read the following:

Martilli, A., Roth, M., Chow, W.T., et al. 2020. Summer average urban-rural surface temperature differences do not indicate the
need for urban heat reduction.

And this article, which they already cite, contains detailed discussion on several issues related to the use of LST that the authors
should consider in my view, but currently do not:

Voigt, J.A. and Oke, T.R., 2003. Thermal remote sensing of urban climates. *Remote sensing of environment*, 86(3), pp.370-384.

In other words, while satellite-derived LST is readily available and therefore often used in these kinds of analyses, I don't think
it is the appropriate data to answer the scientific questions posed here. The link between LST and actual exposure of urban
residents is extremely tenuous at best. Moreover, for assessment of impacts of tree, in particular, it is not appropriate. The
major impact of trees in terms of heat exposure is shade, and LST cannot capture the impacts of shade.

We are grateful to Reviewer 1 for bringing this up. We concur with the reviewer that basing the definition of heat exposure
solely on satellite data can be limited at specific spatiotemporal scales. In the revised manuscript, we added a subsection in
the Introduction where we discussed the limitations of using LST and emphasized that at the scale of our analysis, there is a
clear relationship between air temperature and remotely-sensed surface temperature^{3,4}. It's important to note that our study

focuses on global patterns and seasonal or longer timescales, not hourly/daily conditions at specific locations where differences
between air and surface temperatures could potentially be significant due to weather-scale processes, as noted by Martilli et al.
(2020)². Additionally, our work does not focus on the definition of Surface Urban Heat Island (the difference between rural and
urban areas), but rather on the actual values of LST without any relative comparison. This is because, as shown in Figure 1 of
the manuscript, we do not believe that this definition is representative of heat stress in cities, but rather reflects rural vegetation.
The criticisms raised by Martilli et al. (2020)² and Stewart et al. (2021)⁵ are based on the surface heat island definition, which
is not part of this study. It is worth noting that currently, it is not feasible to obtain daily global information on air temperature
at the spatial scale of our work (i.e. 1 km) for inter-city comparisons.

We also agree on the fact that “thermoscapes” based only on 2-D surface temperatures is a limited definition and we
removed this term from the manuscript: the term was used because that was the aspiration and while we did not reach it we are
building research to go towards it.

At the same time, while the major impact of trees and other plants in terms of heat exposure is shade⁶, we would like also to
emphasize the effect of evapotranspiration⁷. Additionally, vegetation can absorb and store heat, which can further decrease the
temperature in an urban area⁸. Furthermore, trees and vegetation can also reduce the amount of heat absorbed by buildings
and pavement surfaces, which can lower the temperature inside buildings as well⁹. Additionally, the cooling effects of urban
plants go beyond just providing shade for roads. One major way they help is by releasing water through evaporation, which
can absorb a significant amount of net radiation. By lowering the ratio of sensible heat to latent heat (known as the Bowen
ratio), plants can effectively lower air temperature and help combat climate change even in areas that aren't directly shaded.
This can also have an indirect impact on nearby areas due to heat advection. Our data analysis shows high values of the spatial
autocorrelation of LST that is driven by these effects.

Based on the revision we made according to the Reviewer's comments, we believe that the results and conclusions of our
manuscript are robust, correct given the scale and aim of the study, and congruent with existing literature on urban climate and
city analytics.

121 INTRODUCTION

We agree on the different points commented on by Reviewer 1: we addressed all of those points in the Introduction of the
revised manuscript.

• First paragraph:

- – Cities typically have lower albedos than rural areas. Some typical urban materials often have lower albedos, e.g.
asphalt, and many roofing types.
- – We revised this point in the paragraph.
- – Cities typically have lower albedos than rural areas. Some typical urban materials often have lower albedos, e.g.
asphalt, and many roofing types.
- – We revised this point in the paragraph emphasizing the albedo's properties of urban landscapes.
- – Frequency of extreme heat is mostly controlled by synoptic weather patterns, not by the composition of the urban
landscape.
- – We revised this point in the paragraph emphasizing the interplay between climate events and urban landscape.

• Third paragraph:

- – LST does not effectively capture the effects of shade from trees, since shade reduces temperatures of surfaces under
trees that are not accessible to satellite-based remote sensors; therefore, it is not clear why the authors discuss tree
shade.
- – We agreed with Reviewer 1 on this point and we, therefore, revised this part in the Introduction. We should also
mention that shade depends on the acquisition time. As we use Modis Terra which scans at 10:30 am in the morning,
the shadows are partly lateral and could be “seen” by the sensor. Arguably not so much for many trees if they are in
groups, but it would be the case for shades from buildings in urban canyons at some points. We believe that this
point should be studied and analyzed in further studies.

METHODS

We thank Reviewer 1 for the comments. We addressed them below

- Distance to water bodies: the distance will be much more relevant if the water is upwind (for the dominant wind direction) than downwind; I presume this is not taken into account?
- yes, we did not consider the wind direction and we emphasized this point in the revised manuscript.
- Time of day is not reported – what time of day are the results representative of? This should be provided in the Results section as context. How well is the day-to-day temporal variation over the summertime sampled?
- In this research we make use of the MOD11A1 Version 6 product that provides daily per-pixel Land Surface Temperature and Emissivity with 1 kilometer (km) spatial resolution in a 1,200 by 1,200 km grid. The acquisition time of the MOD11A1 Version 6 is around 10:30am and 10:30pm respectively for day and night.

RESULTS

- Climate and population exposure to extreme heat
 - I disagree that “exposure of the urban population to extreme heat” is calculated. It is the co-location of high LST with population. High roof temperature, a primary contributor to LST, is unlikely to impact the population except perhaps those living on the top floor, depending on building construction. Also, the “warmest areas of the city” may be quite different if air temperature was the metric. Finally, “extreme heat” is usually used to indicate synoptically-driven heat (e.g. heat waves), or at very least experiencing (air) temperature exceeding 90th+ percentile of the *temporal* distribution of temperature. The 95th percentile of temperature spatially is not very relevant if nowhere in the city is too hot.
 - We really thank Reviewer 1 for this point. Thanks to his comment we revised the definition of *extreme heat areas* and therefore the *exposure of the urban population to extreme heat*. We aligned our definition with traditional ones that defined the urban population exposure to heat as the number of days per year that exceed a heat exposure threshold multiplied by the total urban population exposed^{1,10}. We measure exposure in person-days/year-1 which is a widely used metric to compare and contrast exposure to extreme heat across geographies and time periods¹¹. We put a lot of effort into this point: in particular, for each city, we calculated the 90th+ percentile of the temporal distribution of LST (LST_{90}) over 20 years of observations (2000-2021) as recommended by Reviewer 1. We computed the thresholds information both for day (LST_{90}^d) and night time (LST_{90}^n) for all the cities and their distribution as shown Figure Rev1 and in Figure Rev2. Then, for each pixel, we calculated the number of days that we observed $LST \geq LST_{90}$ both for days and nights as shown in Figure Rev3 for the city of Paris. For each year, for each pixel of each city, it is possible to count the number of days over the thresholds and also the population. The population data from the GHSL is available only for the years 2010, 2015, and 2020: in order to estimate the population data for all the years we computed a stepwise linear regression. In Figure Rev4 we show the summary of the definition of the total exposure for daytime and nighttime for the city of Paris. In the same way as previous research^{1,10}, we measure exposure in person-days/year-1 as the sum of population multiplied by the days (or nights) over the thresholds per pixel per year.

The new definition of total exposure is:

$$TE = \frac{TE_D + TE_N}{2} \quad (1)$$

where we defined the total exposure (TE) as the average of the exposures between day (TE_D) and night (TE_N). In particular, TE_D and TE_N are respectively the average number (in the 10 years of observations) of days and nights for each pixel times the average population (i.e. the population in 2015) over the thresholds. In the revised version we proposed two spatial regression models that are able to predict with high accuracy the number of days and nights over the thresholds and then define the total exposure.

- • Reduction of the population exposed to extreme heat.
- – What exactly is meant by “the population exposed to the warmest areas of the urban environments can be reduced
by 50%”? Does it mean, for the example of Paris (threshold of 34.4 C), that temperature is reduced to 34.35
C? The issue with the use of percentages above or below a threshold is that a small reduction in temperature that
nevertheless crosses a defined threshold (e.g., from 34.5 C to 34.3 C for Paris) is not that relevant or impactful.
Even the significance of a 1 C LST reduction for the population is not clear, both in terms of the relevance of this
drop in LST and the relevance of LST to actual exposure to heat of urban residents as mentioned above. (Also note
that sometimes Kelvin are used, and sometimes degrees C – suggest more consistency.)
- – We agreed with Reviewer 1 and we changed the definition and the calculation of exposure of the urban population
to extreme heat. In this revised version, we used the definition of the population exposed to heat temperatures
as defined in Equation 1 of this revision. For each city, we compute the reduction of the population exposed by
increasing the value of NDVI.
- – What is the meaning of the bar plots (e.g. Fig. 4C) by climate zone. Are these results representative of all cities in
each climate zone? E.g., are the chosen cities representative of all cities in that climate zone? This is perhaps a
point for discussion.
- – In the revised version we show new results and figures according to the new definition of exposure Temperature
reduction
- – Figs. 5B and S14 are unclear and would benefit from more explanation. Why is the Local vegetation reducing?
- – In the revised version we show new results and figures according to the new definition of exposure

CONCLUSIONS - Delete the final 2 sentences.

We deleted the final 2 sentences.

**Reviewer 2**

This study explores the potential exposure of urban populations to extreme heat under different greening scenarios. It relies on
an interesting approach of combining data about urban population and satellite-derived LSTs. The authors claim that increasing
the overall urban vegetation by 3% can already lead to reduction in exposure to extreme heat of 50%. I think that this statement
may be technically correct for a certain definition of exposure. However, I think that this definition of exposure may not be that
useful when trying to better understand how to protect the urban population from the harmful effects of heat in urban areas.
I also think that the results are presented in a way that suggests an easy solution to reducing the detrimental effects of heat
extremes in cities, but the effects of vegetation in cities are not that clear and easily quantifiable, especially when looking more
holistically at how the effects of vegetation vary over time (e.g. diurnal cycle) and space. I think that the manuscript may in
its current form also lead to misinterpretations of the potential benefits of climate change mitigation (i.e. atmospheric CO2
reduction) in comparison to climate adaptation (i.e. more vegetation in cities). I am suggesting major revisions to improve the
current version of the manuscript.

We thank Reviewer 2 for appreciating our work and for his very relevant suggestions. We revised the manuscript accordingly
and in particular i) we revised our definition of exposure of the urban population to extreme heat and most of the statements
emphasize above, ii) we analyzed both the day and night LST from MOD11A1 and iii) we critically assessed our assumptions
about the effects of vegetation in cities. We tried to revise all the comments made the Reviewer 2 which helped us to really
improve the previous version of the manuscript.

**Major comments:**

Before mentioning my main concerns, I have to say that I am not 100% sure how exposure to extreme heat and the reduction of
it is defined. I think I could not explicitly find it in the methods nor in the Supplementary Information. If I have not missed it, I
would like to ask the authors to specify this more clearly in the manuscript. My understanding is (based on the first paragraph
of the results and Figure S2) that exposure to extreme heat occurs (as per this definition) when LST is higher than the 95th
percentile (as defined within one city) based on average(?) LSTs and I am assuming that exposure is “reduced” whenever
temperatures fall below the temperature for the 95th percentile as predicted by the statistical model for an increase in NDVI. If
I am correct about the definition of exposure, I see certain limitations regarding exposure and the potential to reduce it for the
urban population when increasing vegetation cover (NDVI). I think these limitations should be discussed in the manuscript
and I would suggest to make a less generalized claim than saying that an increase of 3% of vegetation can reduce exposure to
extreme heat by 50%.

We thank Reviewer 2 for emphasizing this point: this comment is very similar to one of the comments raised by Reviewer 1.
We recognized that the previous definition of the exposure of the urban population to extreme heat was not really clear. For this
reason, according also to the comments made by the other Reviewers we revised our definition and therefore we revised our
results and our statements. We revised the definition of *extreme heat areas* and therefore the *exposure of the urban population*
*to extreme heat*. We aligned our definition with traditional ones that defined the urban population exposure to extreme heat as
the number of days per year that exceed a heat exposure threshold multiplied by the total urban population exposed^{1,10}. Let us
say that we put a lot of effort on this point: in particular for each city we calculated the 90th+ percentile of the *temporal*
distribution of LST (LST_{90}) over 20 years.

1. There is now quite an extensive body of literature and controversy about the usefulness of LST in studies about the urban
heat island and the reduction of urban heat (e.g. Martilli et al. 2020). While I would not say that LST data is useless,
I think that we should be aware of certain limitations when defining exposure based on LST: LST can only serve as a
proxy for air temperatures and e.g. human thermal comfort. For example, during the day the canopy urban heat island
(2m air temperature difference) is often quite small (Oke et al., 2017). One explanation seems to be (simply put) that the
city shapes it’s own atmospheric boundary layer in such a way that heat can be transported more efficiently into higher
layers of the atmosphere. This seems to be one of the reasons why air temperatures (during the day) over heavily sealed
and high-rise building areas are sometimes not much higher than over vegetated land outside of the city. This can be
problematic for two reasons: (1) The spatial pattern of the canopy urban heat island (i.e. air temperatures) may overall
(but oftentimes not) correspond to the pattern of LSTs. This means the areas exposed to the highest LST (>95th) may not
always be the ones exposed to the highest air temperatures. I think this should be discussed, since we may argue that
air temperature matters often much more. (2) While the LST distribution ranges between, e.g., 25 and 35 °C, the air
temperature in cities during the day is often not varying spatially by more than 0-2°C. Assuming that a heatwave in a
changing climate may be, e.g., 2°C hotter (compared to some baseline), this may mean that the exposure to extreme heat
and the 95th percentile should have a much broader definition than it has now (including a time dimension)? I think
that this should be discussed and I think it would be great to have a much broader discussion in the paper about why the

author's think that their definition of exposure is useful and not merely arbitrary. For example, why is exposure defined per city and not globally, why not with a fix temperature threshold etc.

We are grateful to Reviewer 2 for bringing up the point about the limitations of using only satellite data to define heat exposure. We concur that this method is limited in terms of spatio-temporal scales. In the revised manuscript, we added a section in the Introduction to discuss the limitations of using land surface temperature (LST) data, but we also emphasized that at the scale of our analysis, there is a clear relationship between air temperature and remotely-sensed LST, as previously reported in the literature^{3,4}. It is important to note that our study focuses on global patterns and seasonal or longer timescales and not on hourly or daily conditions at specific locations, where differences between air and surface temperatures may be more pronounced due to weather-scale processes as emphasized by². Additionally, our study does not focus on the definition of the Surface Urban Heat Island as the difference between rural and urban areas, but rather on the actual values of LST without any relative comparison. We acknowledge that this definition is not representative of heat stress in cities and mainly reflects rural vegetation (see Figure 1 of the manuscript). We discussed those points in a dedicated subsection in the Introduction of the revised manuscript.

We also agree areas exposed to the highest LST (>95th) may not always be the ones exposed to the highest air temperatures: in the revised version of the manuscript, we provide a new definition of exposure to heat that is not a function of the highest LST.

In regard to the last statement, it is important to note that our study uses a global model with the same parameters for all cities. However, we have decided not to use a fixed temperature threshold that is the same for all cities because it would be difficult to compare different cities around the world. For example, a temperature of 30 degrees Celsius in the summer would have a different impact in cities such as Cairo or Toronto. There is a significant amount of literature on the definition of extreme heat for comparison between different areas¹². The threshold for what is considered extreme can be determined in various ways, and the method chosen depends on the research goal. A simple solution is to use a constant absolute threshold, which may be linked to impacts. However, temperature thresholds determined in this way are only relevant in specific geographic regions and time periods, and their values can change with latitude and climate characteristics, as well as the season¹³. This is why relative thresholds have become popular, particularly those based on the empirical temperature distribution at each location, using percentiles (for example, 10% if using the 90th percentile¹²). This approach allows for the comparison of results obtained in different geographic areas with varying climates and seasons. However, it should be noted that this method assumes that the frequency of occurrences is known and that extremes identified in this way may not necessarily be extreme due to their impact. This method is recommended by the Intergovernmental Panel on Climate Change (IPCC)¹⁴ and the World Meteorological Organization (WMO)¹⁵.

Based on the revision we made according to the Reviewer's comments, we believe that the results and conclusions of our manuscript are robust, correct given the scale and aim of the study, and congruent with existing literature on urban climate and city analytics.

2. The diurnal cycle is also something crucial to consider. My understanding is that the MYD11A1 product has nighttime and daytime temperature in it, but that only daytime was used. During nighttime, the effects of vegetation can of course be quite small. Some studies even suggest that, e.g., trees could cause warming (depending on the urban setting). Exposure to extreme heat in this study is thus very narrowly defined as exposure to extreme heat during the day. However, the exposure to extreme heat during the night is at least equally if not more relevant (for example for human health).

We are grateful to Reviewer 2 for bringing attention to the definition of extreme heat areas and the exposure of the urban population to such heat especially for suggesting to consider the diurnal cycle and to use both the MOD11A1 day and night products. We have revised our definition to align it with traditional ones, which define urban population exposure to heat as the number of days per year that exceed a specific heat exposure threshold multiplied by the total urban population exposed. We use the widely accepted metric of person-days/year-1^{1,10} to compare and contrast exposure to extreme heat across different locations and time periods¹¹.

We have put significant effort into addressing this point by calculating the 90th+ percentile of the temporal distribution of land surface temperature (LST) over a 20-year period (2000-2021) for each city. We also calculated the threshold information for both daytime and nighttime for all cities and presented this information in Figure Rev1 and in Figure Rev2.

For each pixel, we calculated the number of days that we observed LST to be greater than or equal to the 90th percentile for both daytime and nighttime, as shown in a figure for the city of Paris Figure Rev3. For each year and each pixel of each city, we were able to count the number of days over the thresholds and also the population. As population data from

314 the GHSL is only available for the years 2010, 2015, and 2020, we estimated population data for all years by using a
315 stepwise linear regression. In a figure, we summarized the definition of the total exposure for daytime and nighttime for
the city of Paris Figure Rev4. In line with previous research, we measure exposure in person-days/year-1 as the sum of
the population multiplied by the days (or nights) over the thresholds per pixel per year.

Our revised definition of total exposure is the average of the exposures between daytime and nighttime, which we have
calculated using two spatial regression models that predict the number of days and nights over the thresholds with high
accuracy. In particular, the new definition of total exposure is:

$$TE = \frac{TE_D + TE_N}{2} \quad (2)$$

where we defined the total exposure (TE) as the average of the exposures between day (TE_D) and night (TE_N). In
particular, TE_D and TE_N are respectively the average number (in the 10 years of observations) of days and nights for
each pixel times the average population (i.e. the population in 2015) over the thresholds. In the revised version we
proposed two spatial regression models that are able to predict with high accuracy the number of days and nights over the
thresholds and then define the total exposure.

- 3. The areas with the highest population density may often be somewhere in or close to the city center and with a lot of
high-rise buildings. There are two things that come to my mind that should probably be considered and discussed: (1)
While the MODIS satellite/sensor will often see overall high LSTs for these areas, there are some places within these
areas where temperatures are actually low. For example, in deep street canyons there is often a lot of shade. In these
canyons the effect of vegetation is often relatively small (e.g. trees will not provide additional shade). Vegetation may
actually not have that much of an effect here and of course, (especially during the night) trees may reduce cold air flows
into the cities because they are natural obstacles. This may be especially relevant in densely built areas with a small
number of wind corridors. (2) When thinking of exposure to extreme heat, we could think of pedestrians and how they
are exposed to heat. I think it is probably a valid assumption that there are a lot of pedestrians in highly populated areas
and because of this many pedestrians may be exposed to heat (unless they walk in heavily shaded deep canyon streets
etc.). However, another dimension is that people are exposed to heat within buildings. In buildings, there is often air
conditioning. So the number of people that (and during how much time of the day) are exposed to extreme heat, may
depend very much on the availability of air conditioning. There might be strong gradients in air conditioning between
wealthy and poor neighborhoods (from the city center to the outskirts). Even though I am not that familiar with the topics
of exposure and vulnerability, it seems to me that these things would matter and that it may even become an issue of
social justice in which neighborhoods it makes sense to increase vegetation and just looking at population density may
be too simple. In addition, it is of course a huge topic, whether vegetation can even have an influence on temperatures
within buildings and how much vegetation effects are only occurring very close to the vegetation itself. So assuming that
a grid cell would profit (the whole grid cell) from a temperature reduction is often not correct, since only the areas with
vegetation will experience a significant temperature decrease, but the urban areas close to these vegetated areas will profit
much less.

We appreciate the Reviewer's comments on the role of vegetation in cities.

- (a) Moreover, Reviewer 2 rises the important point that vegetation may not always have a significant shading effect
in the most central areas of cities with deep urban canyons. This is indeed true where this type of architecture is
dominant (high buildings and narrow roads). On the other hand, it is important to consider that the cooling effects
of urban plants go beyond the direct shading of roads. An important contribution to mitigation derives from the
evaporation of water from plants that may absorb an important share of the net radiation. By reducing the Bowen
ratio (ratio of sensible heat versus latent heat) plants can effectively reduce air temperature and therefore mitigate
climate also in areas that are not directly shaded. This direct effect of plants of air temperature ultimately affects
also neighboring areas due to phenomena of heat advection. This is evident from the results of our data analysis
which show an important role of the spatial autocorrelation of LST. Vegetation can still play a role in regulating
temperature in street canyons, especially during the night when trees can act as natural barriers that impede cold air
flow into the cities. Indeed, while the major impact of trees and other plants in terms of heat exposure is shade⁶, we
would like also to emphasize the effect of evapotranspiration⁷. Additionally, vegetation can absorb and store heat,
which can further decrease the temperature in an urban area⁸. Furthermore, Trees and vegetation can also reduce
the amount of heat absorbed by buildings and pavement surfaces, which can lower the temperature inside buildings
as well⁹. Finally, plant shading on buildings and roofs (green roofs or walls, urban pavement) reduces the heat
storage in walls and roads, ultimately mitigating the night-time heat island effect. We think that the LST metric
used in our paper summarizes the net effect of these different processes in an effective and robust manner.

(b) The Reviewer is correct that exposure to heat is not limited to the outdoors and that people may also be exposed to
heat within buildings. The availability of air conditioning, as well as the gradient in air conditioning between wealthy
and poor neighborhoods, is a crucial factor to consider when assessing exposure to extreme heat. Additionally,
the Reviewer raises a valid point that the effects of vegetation on building temperatures is a complex topic that
would need to be further studied. The Reviewer also highlights that considering population density alone may not
be enough when thinking about where to increase vegetation and that it is an issue of social justice. Overall, this
comment highlights the need for a multi-faceted approach when thinking about exposure to extreme heat and the
importance of considering the built environment and socio-economic factors when assessing vulnerability. Let us
say that currently we are not capable of doing so, but that such work could be envisaged in future developments of
the methods. It would rely on reliable spatialized socio-economic data, which may not be readily available for all
cities worldwide with the same resolution. We tried to answer this part in the introduction and in the conclusion
sections.

Minor comments (somewhat unordered because there was no line numbering, which makes it really hard to comment):

1. In the first paragraph it says “with higher albedo”. I think it is more often argued that cities have a lower albedo or that it
is very dependent on the building materials deployed when constructing cities in different parts of the world.

We thank the reviewer for this comment: indeed it was a mistake and we changed it.

2. First page, last paragraph: “Changes in the global, . . . , and allergic diseases”. I don’t think that this fits very well here.
Maybe it can be integrated somewhere at the beginning of the paragraph or be removed.

We agreed with Reviewer 2 with this comment and we moved to the previous part of the manuscript.

3. Under the regression model: You say it is a multivariate regression model, but you only have one dependent variable. I
am not sure that this is and should be called multivariate. At least, including multiple independent variables does in my
opinion not make it multivariate.

According to this comment, we modified the text in the manuscript.

4. I would assume that NDVI and NDBI have a strong negative correlation. How does that affect the estimated regression
coefficients? I am missing a bit more information about the structure of the variables (correlation, distribution, etc.) used
and if there could indeed be issues with collinearity.

We thank Reviewer 2 for this comment. Indeed, this information was missing and here and in the Supplementary
Materials of the submitted version we added this analysis. The reviewer is right about the negative correlation between
NDVI and NDBI as shown in Figure Rev5 and in Table 1 with a value of the person correlation of ρ 0.78.

We also report the value of the multicollinearity condition number which is a measure of the degree of multicollinearity
in a multiple regression model. A high condition number indicates a high degree of multicollinearity, which can lead
to unstable parameter estimates and difficulty in interpreting the model. If the value of the multicollinearity condition
number is above 30 it indicates a problem with multicollinearity¹⁶. In our case, the value of the multicollinearity condition
number during the training and validation phases is between 9.8 and 10.8 for all the cross-validation settings as shown
in Figure Rev6.

In the revised version, we chose to use only NDVI and distance to water bodies as predictors because for certain cities
we observed a strong negative correlation between NDBI and NDVI. This simplifies the model without affecting its
performance. Our aim is to demonstrate the impact of vegetation in reducing exposure to extreme heat.

5. In addition, I would find it helpful to see a bit of model diagnostics. For example, how do the residuals look like? Are
they normally distributed (normal q-q), what about homoscedasticity and outliers (e.g., residuals vs leverage)?

We thank Reviewer 2 for this comment that allows us to prove that our methodology is valid. We analyzed the distribution
of the residuals in the test set. In Figure Rev7 we show the difference between the OLS and SLM in predicted the
observed values during the test phase. While the OLS is not able to predict the observed values, the SLM seems to
underestimate the greater values. This is confirmed by the distribution of the residuals computed as observed values
minus predicted values as shown in Figure Rev8. From the analysis suggested by Reviewer 2 we can demonstrate
that only the 1% of the observed values can be considered outliers from an accurate analysis of the residuals: we show
in Figure Rev9 in the studentized residuals versus leverage plot.

6. I have to say that I find the statistical approach overall a bit (maybe sometimes unnecessarily) complicated and sometimes the benefits are not entirely clear to me: a) The authors are saying that: “Our analysis shows that the model performs significantly better than a standard linear regression model, underlying the importance of the spatial dimension”. They also claim that they don’t use any “additional climate information”. I consider both statements rather misleading (but of course I may also not fully understand everything). (1) The fact that there is a large improvement when using a “spatial model” seems due to the fact that you include some form of “auto-covariate” which is constructed based on the temperature itself. In my opinion this is “additional climate information” even though it happens to be the dependent variable in the model. (2) I believe that spatial autocorrelation when analyzing LST does not matter that much. I would assume that the large performance improvement in comparison to a “standard regression” approach, would also occur, if you simply include average LST per city as an additional independent variable. I also doubt that accounting for auto-correlation is most important in terms of “performance” (R^2). Usually, I would consider it relevant to account for autocorrelation when you do some sort of statistical inference including the calculation of p-values and to a certain degree it can of course influence your regression coefficients. So to me it would make more sense to show these things (change in p-value and regression coefficients) instead of all the figures illustrating the performance. b) I am assuming that you are using the cross-validation approach to find an optimal “rho” value (which is a hyperparameter that controls, e.g., overfitting?), but I am not entirely sure if this is described explicitly and if my understanding is correct. Wouldn’t it make sense to do the whole cross-validation exercise to find an optimal rho, then estimate the regression coefficients (for a combined training and validation set) and then evaluate the error with the help of the test set? I would say that averaging the regression coefficients (betas) should not have any benefits. Do you systematically vary “rho” to find its best value? If not and if “rho” is fitted as a parameter of the model, then I would again say that the whole cross-validation approach that you are employing does maybe not make much sense because the parameters will probably not be any better than without the cross-validation exercise.

We thank Reviewer 2 for this comment that allows us to prove that our methodology is valid.

- 1) We agree on the fact that somehow we add a sort of climate information in the model. We revised this part in the manuscript and we removed the sentence. Our goal was to explain that we used only remote sensing data without any other information like data from climate background.
- 2) Here we show the values of the performance of a linear regression model in terms of R^2 if we include average LST per city in Figure Rev10. We can see that the performances of the linear regression models are very low even if we consider the average LST (OLS_1) compared to the spatial ones, especially in the phase of cross-validation where we get negative values of the R^2 .
- 3) We agree with Reviewer 2 that the main goal of spatial models is not their performance but rather that they are able to explain autocorrelation. We report the value of the Moran’s I coefficient in Figure Rev11. The analysis shows very high values of the coefficients $I > 0.9$ and this means that is important to account for spatial models in order to provide a better solution because there is a spatial dependence between the dependent and independent variables. Indeed, spatial regression models account for this dependence to provide more accurate results¹⁷. This way we show the value of the performance of the models in order to show the improvement of the accuracy where we account for the autocorrelation of the variables. We do not report the values of the $p - values$ of the coefficients because they are always significant and always below 10^{-7} .
- 4) Regarding the cross-validation part, we probably did not explain well the modus operandi we adopted. Let us say that for an individual observation, the spatially lagged equation can be written as the following:

$$y_i = \sum_{q=1}^Q \beta_q X_{iq} + \rho \sum_{j=1}^n W_{ij} y_j + \varepsilon_i \quad (3)$$

with $j \neq i$. Since the dependent variable, y appears on both sides of the expression:

$$Y = X\beta + \rho WY + \varepsilon \quad (4)$$

we can re-arrange this expression to solve for Y :

$$Y = (I - \rho W)^{-1} X\beta + (I - \rho W)^{-1} \varepsilon \quad (5)$$

From the entire dataset that contains 200 cities, we select 5 sub-datasets containing 40 (10 for each climate zones) cities each corresponding to the 20% of the initial one. In this way, we define 5 test sets. For all the 5 test sets

define 5 training/validation datasets: in Figure Rev12 we show the example for the case of first test tests. Each
training validation dataset is composed of 160 cities (40 for each climate zones): we use this dataset to perform a
k-fold cross-validation with $k = 5$. In each training/validation phase, we use 120 cities during the training where
we estimated the values of the coefficients β that we validate in the validation set. Finally, we choose the average
values of the coefficients corresponding to the best R^2 in the validation phases. Therefore, ρ is not a hyperparameter
of our models. We used the cross-validation settings in order to check the validity of the model under different
circumstances and with different datasets and to estimate the coefficients β that better predict our variables (i.e.
number of days and nights over the thresholds).

7. I guess the assumption that the NDVI has the same effect on temperature in different cities is more or less valid. On the
other hand, I would assume that vegetation in dry regions use less water for transpiration and hence cause less cooling.
Would a discussion on this make sense and did you check how much the regression coefficient of NDVI might vary
across different cities?

The point raised by the Reviewer 2 is really important and should be analyzed in the context of specific models for a
single city, where our approach could applied in terms of geographical weighted regression models. In this research we
use a global approach and the same value of the coefficients for all the cities.

8. In Figure 10 I would find it more helpful to see boxplots in the left part. The right part of the figure is totally unclear to
me. Probably the figure caption should be more explanatory.

In the revised version we have new results according to the new definition of exposure.

9. In the methods “Application of the model and evaluation of the mitigation strategy”. The title of this section/paragraph
should change. There is a lot of description about the cross-validation procedure etc. This could be an additional
paragraph or be included under “The spatial model”. We thank the Reviewer 2 for this comment and we changed the text
accordingly.

References Martilli et al. 2020. Summer average urban-rural surface temperature differences do not indicate the need for urban
heat reduction Oke et al. 2017. Urban Climates.

We added this reference and we discussed it.

Reviewer 3

Thank you for the opportunity to review this manuscript. I applaud the authors for their selection of cities across the world,
both Global North and South, and the inclusion of diverse geographies and climates. While the manuscript's methods are
clearly presented, the overall framing of the manuscript and interpretation of results should be strengthened. Most critically,
the manuscript currently frames land surface temperature (LST) as essentially equivalent to heat exposure. While the authors
could argue LST can be used as a proxy of sorts for heat exposure, they must acknowledge the well-documented limitations
of LST. This framing also influences the interpretation of the results, which need much more nuanced than currently written,
particularly if the paper is to be relevant to and clearly understood by decisionmakers. I offer several overarching comments
first, followed by more specific line feedback.

We thank the Reviewer 3 for appreciating our work for his relevant comments that helped us to significantly improve the
manuscript. We would like to mention that after his positive comment we improved the choice of the cities: in the revised
version we choose 200 cities distributed equally in each climate zones (50 cities in each climate zones). We also considered
this point to improve the cross validation settings in order to consider the climate zones in the training/validation and test
sets. In particular, from the entire dataset that contains 200 cities we select 5 sub-datasets containing 40 (10 for each climate
zones) cities each corresponding to the 20% of the initial one. In this way we define 5 test sets. For all the 5 test sets define
5 training/validation datasets: in Figure Rev12 we show the example for the case of first test tests. Each training validation
dataset is composed by 160 cities (40 for each climate zones): we use this dataset to perform a k-fold cross validation with $k = 5$.

While, regarding the part of LST we added a subsection in the introduction to discuss the limitations and advantages of LST
to study the exposure of urban population to extreme heat. We also changed our definition of exposure to extreme heat.
Performance of the linear regression model with information of LST./

Overarching areas of improvement:

- • As stated above, a more nuanced discussion is needed in the introduction on the drawbacks and limitations of LST and
the evidence of its relationship to heat exposure of community members. Community members have different heat
exposure over time depending on the quality of housing, travel modes, and thermal safety at work or school, etc. If LST
is to be used as a proxy for heat exposure, the authors should acknowledge the limitations and drawbacks of doing so.
For instance, Krayenhoff & Voogt (2010) (below), argue that decisionmakers should aim to reduce the impacts of urban
heat, not the urban heat island itself. Please refer to literature such as:
 - – Martilli, A., Krayenhoff, E.S., & Nazarian, N. (2020). Is the Urban Heat Island intensity relevant for heat mitigation
studies? *Urban Climate*, 31, 100541. <https://doi.org/10.1016/j.uclim.2019.100541>
 - – Krayenhoff, E. S., & Voogt, J. A. (2010). Impacts of urban albedo increase on local air temperature at daily–annual
time scales: model results and synthesis of previous work. *Journal of Applied Meteorology and Climatology*, 49(8),
1634-1648.
 - – Kelly Turner, V., Rogers, M. L., Zhang, Y., Middel, A., Schneider, F. A., Ocón, J. P., ... & Dialesandro, J. (2022).
More than surface temperature: mitigating thermal exposure in hyper-local land system. *Journal of Land Use
Science*, 17(1), 79-99.

We thank the Reviewer 3 for this comment that helped us to really improve the manuscript. We added a subsection in the
introduction to discuss the limitations and advantages of LST to study the exposure of urban population to extreme heat.
We also changed our definition of exposure to extreme heat.

- • In the introduction and discussion, I also recommend acknowledging other types of heat mitigation strategies available
to decisionmakers to limit urban heat including cool/higher albedo materials and surfaces, urban design choices that
can improve thermal comfort in microclimates, and the reduction of waste heat sources through the improved energy
efficiency of buildings and indoor cooling as well as the reduction of gasoline-powered vehicles.

We thank the reviewer: we agreed with him/her and we added this part to the introduction

- • Along the same lines, the authors should be more nuanced in the discussion of urban greening strategies for heat
mitigation, and mention the critical water resource tradeoff decisions that would need to be made in many drought-
stricken areas of the world (particularly arid regions which they identify would need greater amounts of urban greening
to reduce LST).

We thank the reviewer: we agreed with him/her and we added this part to the introduction.

• Finally, the authors should address that it is not simply a matter of increasing the total amount of urban greening in a city,
but where the greening is targeted. Gober et al. (2009) found that irrigated landscapes in an arid region lowered nighttime
temperatures but only in the least vegetated neighborhoods. Similarly, there are numerous studies published about the
relationship between LST and historically marginalized areas within cities (additional references below), so targeting
interventions to those most vulnerable community members should also be considered. While the marginalized areas of
cities are not part of your study, the inequity of where vegetation is most often already located should be addressed in the
introduction and the interpretation of the results.

– Gober, P., Brazel, A., Quay, R., Myint, S., Grossman-Clarke, S., Miller, A., & Rossi, S. (2009). Using watered
landscapes to manipulate urban heat island effects: how much water will it take to cool Phoenix?. *Journal of the*
*American Planning Association*, 76(1), 109-121.

– Locke, D. H., Hall, B., Grove, J. M., Pickett, S. T., Ogden, L. A., Aoki, C., ... & O'Neil-Dunne, J. P. (2021).
Residential housing segregation and urban tree canopy in 37 US Cities. *npj urban sustainability*, 1(1), 1-9.

– Hoffman, J. S., Shandas, V., & Pendleton, N. (2020). The effects of historical housing policies on resident exposure
to intra-urban heat: a study of 108 US urban areas. *Climate*, 8(1), 12.

– Dialesandro, J., Brazil, N., Wheeler, S., & Abunnasr, Y. (2021). Dimensions of thermal inequity: neighborhood
social demographics and urban heat in the Southwestern US. *International journal of environmental research and*
*public health*, 18(3), 941.

We thank the reviewer: we agreed with him/her and we tried to answer to this part in the introduction and in the conclusion
sections.

Specific line feedback:

• Title: The manuscript is only focused on “urban greening strategies” versus a broader set of heat mitigation strategies,
and the focus of the methods is on LST and the reduction of LST, rather than reduction of heat exposure.

We thank the Reviewer 3 on this point and we change the title focusing on the greening intervention strategy.

• Page 1, 2nd paragraph: “As a consequence, heat waves in urban climates will have a profound impact on humankind
as the climate warms up.” Please reflect upon the tense in which heat waves is framed in this sentence. Heat waves are
already having an impact on individuals, a risk that is increasing (versus a risk that is only in the future).

We thank the Reviewer 3 on this point and we changed this part on the manuscript.

• Page 2, Paragraph 1: “To cope with the increasing need to foster climate mitigation and adaptation, it is therefore
imperative to have spatially detailed, temporally sub-daily information for most, if not all cities.” Not sure why this
distinction is being made here, what cities do not need to cope with climate change? Recommend rephrasing to something
like, “. . . information for cities.”

We thank the Reviewer 3 on this point and we changed this part on the manuscript.

• Page 2, Paragraph 1: “. . . subject focused on the so-called Urban Heat Island (UHI) phenomena. . .” Recommend
removing “so-called”.

We thank the Reviewer 3 on this point and we changed this part on the manuscript.

• Page 3, Paragraph 1: “city thermoscape. . .” recommend continuing to use “urban thermoscape” as indicated in the
introduction.

We thank the Reviewer 3 on this point and we removed the term *thermoscape* in the manuscript that could lead to
misinterpretations.

• Page 3, Paragraph 1: “For instance, these plans could be implemented by favouring green spaces and constructing
green or cool roofs and cool pavements.” This statement is correct, but in the context of the paragraph the focus is
urban greening strategies, of which cool roofs and cool pavements are not part of. Recommend rephrasing to clarify the
distinction or removing these strategies from this sentence.

We thank the Reviewer 3 on this point and we removed this part on the manuscript.

• Page 4, Paragraph 1: “However, we are still missing dedicated modelling tools to facilitate the design of city-specific
plans based on the magnitude of the intervention required to reach climate targets.” Can you provide evidence from
the literature that modelling tools are needed, and not training of professionals on existing modelling and information
decision support tools?

- • Page 4, Paragraph 2: Noting it is being referred to as “city thermoscapes” again here.
As we removed the term thermoscape from the manuscript, we would like to say our goal would be to provide in future a
tool or methodology that is able to define a thermoscape of a city. In this research we hope to provide an initial approach
to this.
- • Page 4, Paragraph 3: “It is evident that people in arid regions are located in areas of their particular city less exposed to
extreme heat. . .” Please clarify and reconsider your interpretation of this result. The populations in arid regions may be
located in areas of a city that have lower UHI relative to surrounding natural landscapes, but that does not mean they are
less exposed to extreme heat.
- We are completely agree with this point that it is the reason why we do not want to study UHI bu the real value of heat
for each pixel (without any difference to rural urban areas). Our goal is to model and predict exposure to extreme heat.
We better explained this point in order to make sure that the reader could get the point.
- • Page 5, Paragraph 3: “we show the amount of vegetation we need to increase in order to reduce the exposure of the urban
population to extreme heat areas. . .” More accurate to say that the scenario shows vegetation needed to reduce the LST,
which you are then using as a proxy for exposure of the urban population to heat.
- In this revised version we show different results and we changed the text accordingly.
- • Page 5, Paragraph 3: “Whether this expansion of urban vegetation is actually feasible in that specific area without
compromising the capacity to host people should be further investigated.” Not only the capacity of the area to host people,
but also trade-offs with water use, maintenance costs of the additional vegetation, etc.
- We really agreed with the Reviewer 3 on this point and we added this reflection on the manuscript.
- • Page 6, Paragraph 1: Although the study is focused on urban greening strategies, this would be another good place to nod
to the fact that cities are not only pursuing greening, but have an additional suite of heat mitigation strategies available to
them. As currently written, it sounds like the only option available to cities is urban greening and not a holistic mix of
greening, cool surfaces, and waste heat reduction.
- We really agreed with the Reviewer 3 on this point and we added this reflection on the manuscript. We mention this point
and also the fact that greening is not the only solution but just one of many possible alternatives.
- • Finally, I also recommend including line numbers in future submissions of the manuscript.
We thank the Reviewer 3 on this point and we added the line numbers to the manuscript.

References

- **1.** Tuholske, C. *et al.* Global urban population exposure to extreme heat. *Proc. Natl. Acad. Sci.* **118**, e2024792118, DOI:
[10.1073/pnas.2024792118](https://doi.org/10.1073/pnas.2024792118) (2021).
- **2.** Martilli, A., Roth, M., Chow, W. T. *et al.* Summer average urban-rural surface temperature differences do not indicate the
need for urban heat reduction. (2020).
- **3.** Manoli, G. *et al.* Magnitude of urban heat islands largely explained by climate and population. *Nature* **573**, 55–60 (2019).
- **4.** Zhou, D. *et al.* Satellite remote sensing of surface urban heat islands: Progress, challenges, and perspectives. *Remote. Sens.*
**11**, 48 (2018).
- **5.** Stewart, I. *et al.* Time evolution of the surface urban heat island. *Earth's Futur.* **9**, e2021EF002178 (2021).
- **6.** Armson, D., Stringer, P. & Ennos, A. The effect of tree shade and grass on surface and globe temperatures in an urban area.
*Urban For. & Urban Green.* **11**, 245–255 (2012).
- **7.** Duarte Rocha, A., Vulova, S., van der Tol, C., Förster, M. & Kleinschmit, B. Modelling hourly evapotranspiration in urban
environments with scope using open remote sensing and meteorological data. *Hydrol. earth system sciences* **26**, 1111–1129
(2022).
- **8.** Nuruzzaman, M. Urban heat island: causes, effects and mitigation measures-a review. *Int. J. Environ. Monit. Analysis* **3**,
67–73 (2015).
- **9.** Gago, E. J., Roldan, J., Pacheco-Torres, R. & Ordóñez, J. The city and urban heat islands: A review of strategies to mitigate
adverse effects. *Renew. sustainable energy reviews* **25**, 749–758 (2013).
- **10.** Coffel, E. D., Horton, R. M. & De Sherbinin, A. Temperature and humidity based projections of a rapid rise in global heat
stress exposure during the 21st century. *Environ. Res. Lett.* **13**, 014001 (2017).
- **11.** Matthews, T. K., Wilby, R. L. & Murphy, C. Communicating the deadly consequences of global warming for human heat
stress. *Proc. Natl. Acad. Sci.* **114**, 3861–3866 (2017).
- **12.** Sulikowska, A. & Wypych, A. Summer temperature extremes in europe: how does the definition affect the results? *Theor.*
*Appl. Climatol.* **141**, 19–30 (2020).
- **13.** Stephenson, D. B., Diaz, H. & Murnane, R. Definition, diagnosis, and origin of extreme weather and climate events. *Clim.*
*extremes society* **340**, 11–23 (2008).
- **14.** Field, C. B., Barros, V., Stocker, T. F. & Dahe, Q. *Managing the risks of extreme events and disasters to advance climate*
*change adaptation: special report of the intergovernmental panel on climate change* (Cambridge University Press, 2012).
- **15.** Data, C. Guidelines on analysis of extremes in a changing climate in support of informed decisions for adaptation. *World*
*Meteorol. Organ.* (2009).
- **16.** Weisberg, S. *Applied linear regression*, vol. 528 (John Wiley & Sons, 2005).
- **17.** Anselin, L. & Rey, S. J. Spatial econometrics in an age of CyberGIScience. *Int. J. Geogr. Inf. Sci.* **26**, 2211–2226, DOI:
[10.1080/13658816.2012.664276](https://doi.org/10.1080/13658816.2012.664276) (2012).

Figure Rev1. Boxplot of the dai LST_{90}^d and night LST_{90}^n thresholds.

(a) LST Daily thresholds values: LST_{90}^d

(b) LST Night thresholds values: LST_{90}^n

Figure Rev2. Values of the 90th+ percentile of the temporal distribution of the LST for the different city for the daytime (a) and night time (b).

(a) Number of days over the daily thresholds LST_{90}^d .

(b) Number of nights over the night thresholds LST_{90}^d .

Figure Rev3. Number of days over the critical thresholds for the city of Paris.

Figure Rev4. Definition of total exposure to extreme heat for the city of Paris over the different years.

Figure Rev5. Correlations between the observed variables.

Table 1. Correlations between the observed variables.

	hot_days	hot_nights	NDVI	NDBI	dist_n
hot_days	1.000000	0.665792	-0.419391	0.312625	-0.034275
hot_nights	0.665792	1.000000	-0.370339	0.248157	0.004655
NDVI	-0.419391	-0.370339	1.000000	-0.785032	-0.120981
NDBI	0.312625	0.248157	-0.785032	1.000000	0.025186
dist_n	-0.034275	0.004655	-0.120981	0.025186	1.000000

Figure Rev6. Multicollinearity condition number values.

Figure Rev7. Observed versus predicted values of the OLS and SLM models.

Figure Rev8. Distribution of the residuals for the SLM day model.

Figure Rev9. Leverage vs. Studentized Residuals.

Figure Rev10. Performance of the linear regression model with information of LST.

Figure Rev11. Values of the Moran's I coefficient in the training validation setting of the spatial models.

Figure Rev12. Schematic representation of the cross-validation phase.

REVIEWER COMMENTS

Reviewer #1 (Remarks to the Author):

Reducing extreme heat exposure in cities through spatial analysis of urban greening by Massaro et al.

Comments on Response to Reviewer 1

The authors have done a reasonable job of revising the manuscript; however, I still take issue with several aspects of the current presentation. While in general the authors have executed a good study, the fundamental methods are not adequate for the purpose, in my view. I think Manoli et al. (2019), being published in such a high profile outlet (i.e. Nature), has unfortunately set back the understanding of heat exposure and heat stress in cities by readers who may not have the scientific context to understand the drawbacks or flaws in their method. I am opposed to any further such publications, and in its current state this submission claims that LST is useful in some practical sense to identify areas of increased urban heat exposure and the benefits of vegetation. Yet, there is little proof that this is so and a lot of clear reasons why we would expect it not to be. I do not want to see another high profile article (e.g. in Nature Communications in this case) conflate LST with the actual variables that impact heat exposure simply because LST is readily available and is therefore the basis of the method chosen.

A primary overall comment I have is that I think the use of the term "heat exposure" is not accurate. LST cannot be linked to heat exposure (indoor or outdoor) in a direct way. The use of this term when working with urban air temperature is more justified (but could still be questioned) because air is well mixed via turbulence, and therefore varies more slowly in space. Surface temperature, by contrast, varies at small spatial scales. Remotely-sensed surface temperature includes rooftop and tree top temperatures which are not of much relevance to outdoor heat exposure at ground level (pedestrians). I strongly recommend that the authors speak of surface temperature instead (and be clear about which surface temperature, e.g. see Stewart et al. 2021, Fig. 8). E.g., the title could be: "Reducing extreme surface temperature in cities through spatial analysis of urban greening" [Note: Manoli et al. (2020) make several comments about surface temperature being more useful than air temperature, which are only partially accurate and highly inaccurate if LST is the "surface temperature" metric they indicate. I suggest caution in using this non-peer reviewed publication by scientists who mostly have little urban climate background to support any claims.]

This brings me to a second point. I don't think that spatial analysis of urban greening will reduce surface temperature (or heat exposure), but urban greening may well do so. Therefore, I suggest a title for this submission that is more accurate, such as: "Spatially-optimized urban greening for reduction of urban land surface temperature".

"We changed the definition of population exposure to extreme heat areas, which are now defined as the number of days when heat exposure surpasses a specific threshold multiplied by the total urban population. This new definition is in line with the current literature."
Yes, although literature that uses this definition (Broadbent et al. 2020; Tuholske et al. 2021) assesses air temperature, not surface temperature as is done here. "Exposure" to air temperature and surface temperature are different (see discussion two paragraphs above).

Differences between LST and air temperature in urban areas can be stark, irrespective of the urban-rural difference (i.e., Martilli et al. 2020 critique the use of LST for urban climate assessments more generally, irrespective of urban-rural differences). Again, I strongly recommend the authors read Stewart et al. (2021), e.g., their Figs. 8 and 10. Here is an excerpt from Martilli

et al. 2020:

"Further, the satellite-derived Ts used in [Manoli et al. 2019] is the surface radiant temperature that represents only a subset of urban surfaces seen by the radiometer, biasing the radiometric Ts toward horizontal and away from vertical surfaces. Therefore, satellite derived urban Ts does not represent the complete surface participating in the energy exchange with the atmosphere. It does not fully capture temperatures at pedestrian levels and includes roof-level temperatures that are of questionable relevance to outdoor heat stress and the associated need for mitigation." (See Stewart et al. 2021, Fig. 8.)

"(ii) offer monitoring and modelling systems of the urban thermal environment (hereafter defined as the urban thermoscape)"

This is very vague. The claim seems to be made that LST can capture the "urban thermal environment" or "urban thermoscape":

"...the design and evaluation of urban adaptation plans should be city-specific and based on the detailed assessment of the city thermoscape. To assess the latter requirements, in this study we focus our attention on the drivers of the urban LST gradients across different climate zones. ..."

I think more precision is required – which specific temperature or combination of temperatures are included in the "urban thermal environment" or "urban thermoscape"? It seems that LST is the answer in this submission. Yet, LST clearly misses several elements of importance to the "urban thermal environment", e.g. building walls, ground underneath trees, and includes others of less consequence (e.g. rooftops, depending on the application) and it ultimately a biased measure of the "urban thermal environment".

Even if the LST-air temperature relationship can be relied on (and there is only partial supporting evidence for this from only one study that Manoli et al. 2020 mention – Zhang et al. 2014 – there is little evidence I can find in the other papers they cite), this does not suggest that the LST differences induced by an intervention such as vegetation (or specifically trees) correlates to their air temperature differences. Trees, for example, affect air temperature in very complex ways (via at least four physical mechanisms – see work by Manoli and colleagues in Meili et al. 2021 among that by other scholars):

Meili, N., Manoli, G., Burlando, P., Carmeliet, J., Chow, W.T., Coutts, A.M., Roth, M., Velasco, E., Vivoni, E.R. and Faticchi, S., 2021. Tree effects on urban microclimate: Diurnal, seasonal, and climatic temperature differences explained by separating radiation, evapotranspiration, and roughness effects. *Urban Forestry & Urban Greening*, 58, p.126970.

"The study of LST allows for a more detailed and global representation of temperature patterns at both fine temporal (daily) and spatial (1km resolution) resolution, thereby enabling inter-city comparisons, that would not be feasible with air temperature data."

The Tuholske et al. (2021) study uses an air temperature dataset that largely contradicts this point.

Is satellite acquisition time reported in the Methods? I cannot find it.

Finally, the English language requires improvement in some of the newly-written sections.

Reviewer #2 (Remarks to the Author):

I thank the reviewers for their effort and I think that they substantially improved the manuscript. They acknowledge several limitations including the ones that stem from relying on LST. For me it would be important that it is clear from the beginning how exposure is defined in this study and

how it should be understood.

This means that the definition in the abstract: "We determine exposure by calculating the number of days when heat exposure surpasses a specific threshold multiplied by the total urban population affected." should be replaced by the definition in the conclusions which is much clearer and much more meaningful "We defined exposure as the number of days per year where LST exceeds a heat exposure threshold multiplied by the total urban population exposed, in person days."

In the abstract it also says: "Here, for the first time we implement a spatial regression model based on remote sensing data that is able to assess, with high accuracy, the population exposure to extreme heat in urban environments across 200 cities based on surface properties like vegetation cover and distance to water bodies." I don't agree with the statement "with high accuracy". This is purely based on the statistical fit as far as I understand. However, it does not pay attention to the fact that LST is a very specific and often inadequate indicator for "heat".

Reviewer #3 (Remarks to the Author):

Thank you for the opportunity to review the revised manuscript. The revisions have greatly improved the study's overall framing and provided more nuance for its potential application in both research and practice. The discussion of the limitations of LST and references to other heat mitigation strategies, and the importance of considering indoor heat exposure, are welcome and better situates this study in current heat literature. My major feedback has been resolved, although I list several minor specific line comments below.

Specific line feedback:

- Line 52 (Figure 1): Consider a simple symbology for both Kyoto and Jeddah (since they are the cases listed) that help the reader identify them quickly on the world map and the graph.
- Line 156 (Figure 4): Recommend briefly describing what Scenario 1 and 2 are in the caption (or giving them a more descriptive short-hand title)
- Line 190 (Figure 5): Recommend clarifying caption for "C" to again make Scenario 1 and 2 more clear, such as: "C) Global NDVI increment in order to achieve the exposure reduction by targeting the entire city (Scenario 1) versus the most populated pixels (Scenario 2), as described in Figure 4"
- Line 217: It may be more inclusive to refer to "The availability of indoor cooling" versus air conditioning, since there are other technologies/non-mechanical options for cooling indoor spaces that may be appropriate (particularly in locations with lack of access to energy)
- Line 265: Refers to Barcelona in Figure S2, but Figure S2 is Guangzhou
- Figure S6: Some odd spelling mistakes, "tresholds" in titles and "botto" in the caption.

Spatially-optimized urban greening for reduction of population exposure to land surface temperature extremes

Response to reviewers

Emanuele Massaro, Gregory Duveiller, Rossano Schifanella, Matteo Piccardo, Luca Caporaso, Hannes Taubenböck, Alessandro Cescatti

Dear Esteemed Reviewers,

We would like to express our sincere gratitude for taking the time to review our manuscript and provide us with valuable feedback. Your constructive comments and criticisms have helped us improve the quality of our research and the manuscript.

We are pleased to inform you that we have taken into account all of your suggestions and made the necessary revisions to the manuscript. Please find our detailed point-by-point response to your comments below, with your original comments highlighted in blue.

The major changes that we have made to the manuscript include:

1. **Indicator change:** we acknowledge that Land Surface Temperature (LST) may not be an adequate indicator for measuring population exposure to heat, and we have revised the definition to consider population exposure to LST extremes instead of population exposure to extreme heat. We have defined exposure as the number of days per year where LST exceeds a given threshold multiplied by the total urban population living in the same pixel, in person-day.
2. **Title change:** we have revised the title of the manuscript as suggested.
3. **Language revision:** we have thoroughly reviewed and revised the manuscript to improve the clarity and readability of the language. We have also addressed all minor comments that you have provided.

Once again, we would like to express our sincere appreciation for your efforts and contribution to this research. We are confident that the revised manuscript will meet your expectations, and we look forward to your feedback.

Sincerely yours,

Emanuele Massaro, Gregory Duveiller, Rossano Schifanella, Matteo Piccardo, Luca Caporaso, Hannes Taubenböck, Alessandro Cescatti

Reviewer 1

The authors have done a reasonable job of revising the manuscript; however, I still take issue with several aspects of the current presentation. While in general the authors have executed a good study, the fundamental methods are not adequate for the purpose, in my view. I think Manoli et al. (2019), being published in such a high profile outlet (i.e. Nature), has unfortunately set back the understanding of heat exposure and heat stress in cities by readers who may not have the scientific context to understand the drawbacks or flaws in their method. I am opposed to any further such publications, and in its current state this submission claims that LST is useful in some practical sense to identify areas of increased urban heat exposure and the benefits of vegetation. Yet, there is little proof that this is so and a lot of clear reasons why we would expect it not to be. I do not want to see another high profile article (e.g. in Nature Communications in this case) conflate LST with the actual variables that impact heat exposure simply because LST is readily available and is therefore the basis of the method chosen.

A primary overall comment I have is that I think the use of the term “heat exposure” is not accurate. LST cannot be linked to heat exposure (indoor or outdoor) in a direct way. The use of this term when working with urban air temperature is more justified (but could still be questioned) because air is well mixed via turbulence, and therefore varies more slowly in space. Surface temperature, by contrast, varies at small spatial scales. Remotely-sensed surface temperature includes rooftop and tree top temperatures which are not of much relevance to outdoor heat exposure at ground level (pedestrians). I strongly recommend that the authors speak of surface temperature instead (and be clear about which surface temperature, e.g. see Stewart et al. 2021, Fig. 8). E.g., the title could be: “Reducing extreme surface temperature in cities through spatial analysis of urban greening” [Note: Manoli et al. (2020) make several comments about surface temperature being more useful than air temperature, which are only partially accurate and highly inaccurate if LST is the “surface temperature” metric they indicate. I suggest caution in using this non-peer reviewed publication by scientists who mostly have little urban climate background to support any claims.] This brings me to a second point. I don’t think that spatial analysis of urban greening will reduce surface temperature (or heat exposure), but urban greening may well do so. Therefore, I suggest a title for this submission that is more accurate, such as: “Spatially-optimized urban greening for reduction of urban land surface temperature”.

“We changed the definition of population exposure to extreme heat areas, which are now defined as the number of days when heat exposure surpasses a specific threshold multiplied by the total urban population. This new definition is in line with the current literature.” Yes, although literature that uses this definition (Broadbent et al. 2020; Tuholske et al. 2021) assesses air temperature, not surface temperature as is done here. “Exposure” to air temperature and surface temperature are different (see discussion two paragraphs above).

Differences between LST and air temperature in urban areas can be stark, irrespective of the urban-rural difference (i.e., Martilli et al. 2020 critique the use of LST for urban climate assessments more generally, irrespective of urban-rural differences). Again, I strongly recommend the authors read Stewart et al. (2021), e.g., their Figs. 8 and 10. Here is an excerpt from Martilli et al. 2020: “Further, the satellite-derived Ts used in [Manoli et al. 2019] is the surface radiant temperature that represents only a subset of urban surfaces seen by the radiometer, biasing the radiometric Ts toward horizontal and away from vertical surfaces. Therefore, satellite derived urban Ts does not represent the complete surface participating in the energy exchange with the atmosphere. It does not fully capture temperatures at pedestrian levels and includes roof-level temperatures that are of questionable relevance to outdoor heat stress and the associated need for mitigation.” (See Stewart et al. 2021, Fig. 8.)

“(ii) offer monitoring and modelling systems of the urban thermal environment (hereafter defined as the urban thermoscape)” This is very vague. The claim seems to be made that LST can capture the “urban thermal environment” or “urban thermoscape”: “... the design and evaluation of urban adaptation plans should be city-specific and based on the detailed assessment of the city thermoscape. To assess the latter requirements, in this study we focus our attention on the drivers of the urban LST gradients across different climate zones. ...” I think more precision is required – which specific temperature or combination of temperatures are included in the “urban thermal environment” or “urban thermoscape”? It seems that LST is the answer in this submission. Yet, LST clearly misses several elements of importance to the “urban thermal environment”, e.g. building walls, ground underneath trees, and includes others of less consequence (e.g. rooftops, depending on the application) and it ultimately a biased measure of the “urban thermal environment”.

Even if the LST-air temperature relationship can be relied on (and there is only partial supporting evidence for this from only one study that Manoli et al. 2020 mention – Zhang et al. 2014 – there is little evidence I can find in the other papers they cite), this does not suggest that the LST differences induced by an intervention such as vegetation (or specifically trees) correlates to their air temperature differences. Trees, for example, affect air temperature in very complex ways (via at least four physical mechanisms – see work by Manoli and colleagues in Meili et al. 2021 among that by other scholars):

Meili, N., Manoli, G., Burlando, P., Carmeliet, J., Chow, W.T., Coutts, A.M., Roth, M., Velasco, E., Vivoni, E.R. and Fatichi, S., 2021. Tree effects on urban microclimate: Diurnal, seasonal, and climatic temperature differences explained by separating

radiation, evapotranspiration, and roughness effects. *Urban Forestry & Urban Greening*, 58, p.126970.

We express our gratitude to Reviewer 1 for acknowledging our efforts and revisions made to the manuscript. The feedback provided regarding the limitation of using LST for heat exposure was taken seriously, resulting in a revised definition of exposure that places greater emphasis on land surface temperature extremes, as reflected in both the title and text. The updated version emphasizes the urban residents residing in regions that encounter a significant number of days surpassing thresholds and are exposed to elevated land surface temperatures (LST).

We changed the title in *Spatially-optimized urban greening for reduction of population exposure to land surface temperature extremes*.

We would like to note that we eliminated the term *thermoscape* from the updated version since we reached a consensus that it was not fitting for this study. Our final goal and hope would be to utilize LST observations along with climate models and other data to establish an accurate definition of an *urban thermoscape* in future studies.

“The study of LST allows for a more detailed and global representation of temperature patterns at both fine temporal (daily) and spatial (1km resolution) resolution, thereby enabling inter-city comparisons, that would not be feasible with air temperature data.” The Tuholske et al. (2021) study uses an air temperature dataset that largely contradicts this point.

We appreciate the Reviewer for raising this point. In the Tuholske et al. (2021) study, the CHIRTS air temperature dataset was utilized, which has a spatial resolution of approximately 0.05° (equivalent to 5km). What we would like to point out in the manuscript is that obtaining measured or modeled air temperature data at a resolution of 1km for any city on a daily basis is currently unattainable.

Is satellite acquisition time reported in the Methods? I cannot find it.

We added this information in the Methods section.

Finally, the English language requires improvement in some of the newly-written sections.

We thank the Reviewer for this point. We revised the manuscript carefully improving the English and fixing all the typos.

Reviewer 2

I thank the reviewers for their effort and I think that they substantially improved the manuscript. They acknowledge several limitations including the ones that stem from relying on LST. For me it would be important that it is clear from the beginning how exposure is defined in this study and how it should be understood. This means that the definition in the abstract: “We determine exposure by calculating the number of days when heat exposure surpasses a specific threshold multiplied by the total urban population affected.” should be replaced by the definition in the conclusions which is much clearer and much more meaningful “We defined exposure as the number of days per year where LST exceeds a heat exposure threshold multiplied by the total urban population exposed, in person days.” In the abstract it also says: “Here, for the first time we implement a spatial regression model based on remote sensing data that is able to assess, with high accuracy, the population exposure to extreme heat in urban environments across 200 cities based on surface properties like vegetation cover and distance to water bodies.” I don’t agree with the statement “with high accuracy”. This is purely based on the statistical fit as far as I understand. However, it does not pay attention to the fact that LST is a very specific and often inadequate indicator for “heat”.

We extend our thanks to Reviewer 2 for recognizing the hard work and improvements made to the manuscript. We report the major changes in the following:

- we added a definition of exposure in the abstract as suggested;
- we removed the claim that we assess the exposure with *high accuracy*;
- we concurred that land surface temperature (LST) is a highly specialized and sometimes insufficient measure for determining *heat*, and as a result, we revised the title and definitions. The updated version concentrates on the exposure of urban residents to elevated LST in areas where there is a substantial number of days and nights surpassing thresholds. In the new version we talk about urban population exposure to LST extremes instead of exposure to *heat*.

Reviewer 3

Thank you for the opportunity to review the revised manuscript. The revisions have greatly improved the study's overall framing and provided more nuance for its potential application in both research and practice. The discussion of the limitations of LST and references to other heat mitigation strategies, and the importance of considering indoor heat exposure, are welcome and better situates this study in current heat literature. My major feedback has been resolved, although I list several minor specific line comments below.

We express our gratitude to Reviewer 3 for acknowledging our efforts and the modifications we implemented. In our subsequent revisions, we have taken into account all of Reviewer 3's feedback.

Specific line feedback:

- Line 52 (Figure 1): Consider a simple symbology for both Kyoto and Jeddah (since they are the cases listed) that help the reader identify them quickly on the world map and the graph.
- We added two arrows with texts to identify the two cities
- Line 156 (Figure 4): Recommend briefly describing what Scenario 1 and 2 are in the caption (or giving them a more descriptive short-hand title)
- We added a description of the scenarios in the caption of the Figure.
- Line 190 (Figure 5): Recommend clarifying caption for “C” to again make Scenario 1 and 2 more clear, such as: “C) Global NDVI increment in order to achieve the exposure reduction by targeting the entire city (Scenario 1) versus the most populated pixels (Scenario 2), as described in Figure 4”
- We added a description of the scenarios in the caption of the Figure.
- Line 217: It may be more inclusive to refer to “The availability of indoor cooling” versus air conditioning, since there are other technologies/non-mechanical options for cooling indoor spaces that may be appropriate (particularly in locations with lack of access to energy)
- We thank Reviewer 3 for this comment and we changed the phrase accordingly.
- Line 265: Refers to Barcelona in Figure S2, but Figure S2 is Guangzhou.
- We made the correction for this reference.
- Figure S6: Some odd spelling mistakes, “tresholds” in titles and “botto” in the caption.
- We really thank Reviewer 3 for noticing those typos that we corrected accordingly.

REVIEWERS' COMMENTS

Reviewer #1 (Remarks to the Author):

The authors have taken the previous set of comments seriously and improved the manuscript. I still think that the word "exposure", e.g. in the title and abstract, is misleading in the context of LST. A significant component of LST, especially extremely hot LST in built up areas, derives from hot roof surface temperatures. However, pedestrians may be experiencing a largely shaded environment between the buildings. So pedestrians are not exposed to LST or at least to significant contributors to LST.

Conversely, I think the authors get it right in the following statement in the "Limitations" section: "For these reasons, in this research we do not focus on the population exposure to heat but to urban areas with high values of LST."

That is, most accurately described, the authors are addressing the co-location of LST variation with population, not the exposure of the population to LST.

Spatially-optimized urban greening for reduction of population exposure to land surface temperature extremes

Response to Reviewer 1

Emanuele Massaro, Gregory Duveiller, Rossano Schifanella, Matteo Piccardo, Luca Caporaso, Hannes Taubenböck, Alessandro Cescatti

Dear Esteemed Reviewer,

Once again, we would like to express our sincere appreciation for your efforts and contribution to this research. We are confident that the revised manuscript will meet your expectations, and we look forward to your feedback.

Sincerely yours,

Emanuele Massaro, Gregory Duveiller, Rossano Schifanella, Matteo Piccardo, Luca Caporaso, Hannes Taubenböck, Alessandro Cescatti

Reviewer 1

The authors have taken the previous set of comments seriously and improved the manuscript. I still think that the word "exposure", e.g. in the title and abstract, is misleading in the context of LST. A significant component of LST, especially extremely hot LST in built up areas, derives from hot roof surface temperatures. However, pedestrians may be experiencing a largely shaded environment between the buildings. So pedestrians are not exposed to LST or at least to significant contributors to LST. Conversely, I think the authors get it right in the following statement in the "Limitations" section: "For these reasons, in this research we do not focus on the population exposure to heat but to urban areas with high values of LST." That is, most accurately described, the authors are addressing the co-location of LST variation with population, not the exposure of the population to LST.

We want to thank Reviewer 1 again for recognizing our hard work in our research. In our study, we refer to the intersection or shared location of two variables, namely LST (derived from satellite) and population, as *exposure*. To clarify, we would say that "variable y is subject to exposure from variable" or "there exists exposure between variables x and y". Essentially, we treat LST as a variable and use the population layer as another variable.

Furthermore, we acknowledge that LST may not always represent ground-level temperatures. However, we argue that high LST affects not only pedestrians but also individuals residing in higher floors of buildings, who experience higher temperatures than those on the ground floor. Therefore, we believe that the term exposure is appropriate for our purposes also in this case.

Finally, we acknowledged and we agreed in the previous revisions about the limitation that LST may not be suitable for calculating heat estimates in the same manner as air temperature. However, we would like to emphasize that we utilize the term *exposure of population to LST extremes* to refer to the presence of both individuals and LST extremes within a given pixel.